# Learning DAGs from Data with Few Root Causes

**Panagiotis Misiakos, Chris Wendler and Markus Püschel**
Department of Computer Science, ETH Zurich
{pmisiakos, wendlerc, markusp}@ethz.ch

## Abstract

We present a novel perspective and algorithm for learning directed acyclic graphs (DAGs) from data generated by a linear structural equation model (SEM). First, we show that a linear SEM can be viewed as a linear transform that, in prior work, computes the data from a dense input vector of random valued root causes (as we will call them) associated with the nodes. Instead, we consider the case of (approximately) few root causes and also introduce noise in the measurement of the data. Intuitively, this means that the DAG data is produced by few data generating events whose effect percolates through the DAG. We prove identifiability in this new setting and show that the true DAG is the global minimizer of the $L^0$-norm of the vector of root causes. For data satisfying the few root causes assumption, we show superior performance compared to prior DAG learning methods.

## 1 Introduction

We consider the problem of learning the edges of an unknown directed acyclic graph (DAG) given data indexed by its nodes. DAGs can represent causal dependencies (edges) between events (nodes) in the sense that an event only depends on its predecessors. Thus, DAG learning has applications in causal discovery, which, however, is a more demanding problem that we do not consider here, since it requires further concepts of causality analysis, in particular interventions [Peters et al., 2017]. However, DAG learning is still NP-hard in general [Chickering et al., 2004]. Hence, in practice, one has to make assumptions on the data generating process, to infer information about the underlying DAG in polynomial time. Recent work on DAG learning has focused on identifiable classes of causal data generating processes. A frequent assumption is that the data follow a structural equation model (SEM) [Shimizu et al., 2006, Zheng et al., 2018, Gao et al., 2021], meaning that the value of every node is computed as a function of the values of its direct parents plus noise. Zheng et al. [2018] introduced NOTEARS, a prominent example of DAG learning, which considers the class of linear SEMs and translates the acyclicity constraint into a continuous form for easier optimization. It inspired subsequent works to use continuous optimization schemes for both linear [Ng et al., 2020] and nonlinear SEMs [Lachapelle et al., 2019, Zheng et al., 2020]. A more expansive discussion of related work is provided in Section 4.

In this paper we also focus on linear SEMs but change the data generation process. We first translate the common representation of a linear SEM as recurrence into an equivalent closed form. In this form, prior data generation can be viewed as linearly transforming an i.i.d. random, dense vector of root causes (as we will call them) associated with the DAG nodes as input, into the actual data on the DAG nodes as output. Then we impose sparsity in the input (few root causes) and introduce measurement noise in the output. Intuitively, this assumption captures the reasonable situation that DAG data may be mainly determined by few data generation events of predecessor nodes that percolate through the DAG as defined by the linear SEM. Note that our use of the term root causes is related to but different from the one in the root cause analysis by Ikram et al. [2022].

**Contributions.** We provide a novel DAG learning method designed for DAG data generated by linear SEMs with the novel assumption of few root causes. Our specific contributions include the following:

37th Conference on Neural Information Processing Systems (NeurIPS 2023).

- We provide an equivalent, closed form of the linear SEM equation showing that it can be viewed as a linear transform obtained by the reflexive-transitive closure of the DAG's adjacency matrix. In this form, prior data generation assumed a dense, random vector as input, which we call root causes.

- In contrast, we assume a sparse input, or few root causes (with noise) that percolate through the DAG to produce the output, whose measurement is also subject to noise. Interestingly, Seifert et al. [2023] identify the assumption of few root causes as a form of Fourier-sparsity.

- We prove identifiability of our proposed setting under weak assumptions, including zero noise in the measurements. We also show that, given enough data, the original DAG is the unique minimizer of the associated optimization problem in the case of absent noise.

- We propose a novel and practical DAG learning algorithm, called SparseRC, for our setting, based on the minimization of the $L^1$-norm of the approximated root causes.

- We benchmark SparseRC against prior DAG learning algorithms, showing significant improvements for synthetic data with few root causes and scalability to thousands of nodes. SparseRC also performs among the best on real data from a gene regulatory network, demonstrating that the assumption of few root causes can be relevant in practice.

## 2 Linear SEMs and Root Causes

We first provide background on prior data generation for directed acyclic graphs (DAGs) via linear SEMs. Then we present a different viewpoint on linear SEMs based on the concept of root causes that we introduce. Based on it, we argue for data generation with few root causes, present it mathematically and motivate it, including with a real-world example.

**DAG.** We consider DAGs $\mathcal{G} = (V, E)$ with $|V| = d$ vertices, $E$ the set of directed edges, no cycles including no self-loops. We assume the vertices to be sorted topologically and set accordingly $V = \{1, 2, ..., d\}$. Further, $a_{ij} \in \mathbb{R}$ is the weight of edge $(i, j) \in E$, and

$$\mathbf{A} = (a_{ij})_{i,j \in V} = \begin{cases} a_{ij}, \text{ if } (i,j) \in E, \\ 0, \text{ else.} \end{cases} \tag{1}$$

is the weighted adjacency matrix. $\mathbf{A}$ is upper triangular with zeros on the diagonal and thus $\mathbf{A}^d = \mathbf{0}$.

**Linear SEM.** Linear SEMs [Peters et al., 2017] formulate a data generating process for DAGs $\mathcal{G}$. First, the values at the sources of a DAG $\mathcal{G}$ are initialized with random noise. Then, the remaining nodes are processed in topological order: the value $x_j$ at node $j$ is assigned the linear combination of its parents' values (called the causes of $x_j$) plus independent noise. Mathematically, a data vector $\mathbf{x} = (x_1, ..., x_d) \in \mathbb{R}^d$ on $\mathcal{G}$ follows a linear SEM if $x_j = \mathbf{x}\mathbf{A}_{:,j} + n_j$, where the subscript $:, j$ denotes column $j$, and $n_j$ are i.i.d. random noise variables [Shimizu et al., 2006, Zheng et al., 2018, Ng et al., 2020]. Assuming $n$ such data vectors collected as the rows of a matrix $\mathbf{X} \in \mathbb{R}^{n \times d}$, and the noise variables in the matrix $\mathbf{N}$, the linear SEM is typically written as

$$\mathbf{X} = \mathbf{X}\mathbf{A} + \mathbf{N}. \tag{2}$$

### 2.1 Transitive closure and root causes

Equation (2) can be viewed as a recurrence for computing the data values $\mathbf{X}$ from $\mathbf{N}$. Here, we interpret linear SEMs differently by first solving this recurrence into a closed form. We define $\overline{\mathbf{A}} = \mathbf{A} + \mathbf{A}^2 + ... + \mathbf{A}^{d-1}$, which is the Floyd-Warshall (FW) transitive closure of $\mathbf{A}$ over the ring $(\mathbb{R}, +, \cdot)$ [Lehmann, 1977], and $\mathbf{I} + \overline{\mathbf{A}}$ the associated reflexive-transitive closure of $\mathbf{A}$. Since $\mathbf{A}^d = \mathbf{0}$ we have $(\mathbf{I} - \mathbf{A})(\mathbf{I} + \overline{\mathbf{A}}) = \mathbf{I}$ and thus can isolate $\mathbf{X}$ in (2):

**Lemma 2.1.** *The linear SEM* (2) *computes data* $\mathbf{X}$ *as*

$$\mathbf{X} = \mathbf{N}\left(\mathbf{I} + \overline{\mathbf{A}}\right). \tag{3}$$

*In words, the data values in* $\mathbf{X}$ *are computed as linear combinations of the noise values* $\mathbf{N}$ *of all predecessor nodes with weights given by the reflexive-transitive closure* $\mathbf{I} + \overline{\mathbf{A}}$.

Since $\mathbf{X}$ is uniquely determined by $\mathbf{N}$, we call the latter the *root causes* of $\mathbf{X}$. The root causes must not be confused with the root nodes (sources) of the DAG, which are the nodes without parents. In particular, $\mathbf{N}$ can be non-zero in nodes that are not sources.

**Few root causes.** In (3) we view $\mathbf{N}$ as the input to the linear SEM at each node and $\mathbf{X}$ as the measured output at each node. With this viewpoint, we argue that it makes sense to consider a data generation process that differs in two ways from (3) and thus (2). First, we assume that only few nodes produce a relevant input that we call $\mathbf{C}$, up to low magnitude noise $\mathbf{N}_c$. Second, we assume that the measurement of $\mathbf{X}$ is subject to noise $\mathbf{N}_x$. Formally, this yields the closed form

$$\mathbf{X} = (\mathbf{C} + \mathbf{N}_c)\left(\mathbf{I} + \overline{\mathbf{A}}\right) + \mathbf{N}_x. \tag{4}$$

Multiplying both sides by $\left(\mathbf{I} + \overline{\mathbf{A}}\right)^{-1} = (\mathbf{I} - \mathbf{A})$ yields the (standard) form as recurrence

$$\mathbf{X} = \mathbf{X}\mathbf{A} + (\mathbf{C} + \mathbf{N}_c) + \mathbf{N}_x\left(\mathbf{I} - \mathbf{A}\right), \tag{5}$$

i.e., $\mathbf{N}$ in (2) is replaced by $\mathbf{C} + \mathbf{N}_c + \mathbf{N}_x\left(\mathbf{I} - \mathbf{A}\right)$, which in general is not i.i.d. Note that (4) generalizes (3), obtained by assuming zero root causes $\mathbf{C} = \mathbf{0}$ and zero measurement noise $\mathbf{N}_x = \mathbf{0}$.

We consider $\mathbf{C}$ not as an additional noise variable but as the actual information, i.e., the relevant input data at each node, which then percolates through the DAG as determined by the SEM to produce the final output data $\mathbf{X}$, whose measurement, as usual in practice, is subject to noise. Few root causes mean that only few nodes input relevant information in one dataset. This assumption can be stated as:

**Definition 2.2** (Few Root Causes assumption). We assume that the input data $(\mathbf{C} + \mathbf{N}_c) + \mathbf{N}_x\left(\mathbf{I} - \mathbf{A}\right)$ to a linear SEM are approximately sparse. Formally this holds if:

$$\|\mathbf{C}\|_0 / nd < \epsilon \quad \text{(few root causes)},$$

$$\frac{\|\mathbf{N}_c + \mathbf{N}_x\left(\mathbf{I} - \mathbf{A}\right)\|_1 / nd}{\|\mathbf{C}\|_1 / \|\mathbf{C}\|_0} < \delta \quad \text{(negligible noise)}, \tag{6}$$

where $\|\cdot\|_0$ counts the nonzero entries of $\mathbf{C}$, $\|\cdot\|_1$ is elementwise, and $\epsilon, \delta$ are small constants.

For example, in Appendix A we show that (6) holds (in expectation) for $\epsilon = \delta = 0.1$ if $\mathbf{N}_c, \mathbf{N}_x$ are normally distributed with zero mean and standard deviation $\sigma = 0.01$ and the ground truth DAG $\mathcal{G}$ has uniformly random weights in $[-1, 1]$ and average degree 4. The latter two conditions bound the amplification of $\mathbf{N}_x$ in (6) when multiplied by $\mathbf{I} - \mathbf{A}$.

The term root cause in the root cause analysis by Ikram et al. [2022] refers to the ancestor that was the only cause of some effect(s). If there is only one root cause in our sense, characterized by a non-zero value, then determining its location is the problem considered by Ikram et al. [2022].

## 2.2 Example: Pollution model

Linear SEMs and associated algorithms have been extensively studied in the literature [Loh and Bühlmann, 2014, Peters and Bühlmann, 2014, Ghoshal and Honorio, 2017, Aragam and Zhou, 2015]. However, we are not aware of any real-world example given in a publication. The motivation for using linear SEMs includes the following two reasons: $d$-variate Gaussian distributions can always be expressed as linear SEMs [Aragam and Zhou, 2015] and linearity is often a workable assumption when approximating non-linear systems.

Our aim is to explain why we propose the assumption of few root causes. The high-level intuition is that it can be reasonable to assume that, with the view point of (3), the relevant DAG data is triggered by sparse events on the input size and not by random noise. To illustrate this, we present a linear SEM that describes the propagation of pollution in a river network.

**DAG.** We assume a DAG describing a river network. The acyclicity is guaranteed since flows only occur downstream. The nodes $i \in V$ represent geographical points of interest, e.g., cities, and edges are rivers connecting them. We assume that the cities can pollute the rivers. An edge weight $a_{ij} \in [0, 1]$, $(i, j) \in E$, captures what fraction of a pollutant inserted at $i$ reaches the neighbour $j$. An example DAG with six nodes is depicted in Fig. 1a.

**Transitive closure.** Fig. 1b shows the transitive closure of the DAG in (a). The $(i, j)$-th entry of the transitive closure is denoted with $\overline{a}_{ij}$ and represents the total fraction of a pollutant at $i$ that reaches $j$ via all connecting paths.

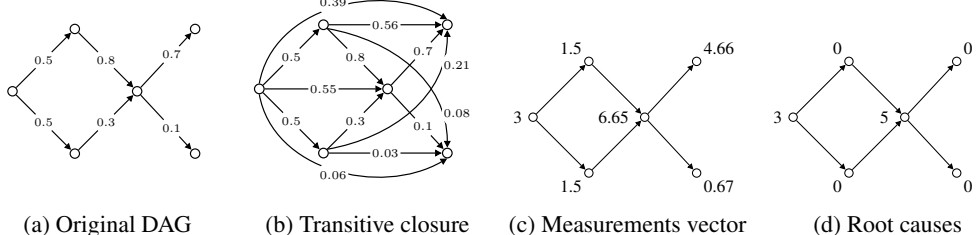

| (a) Original DAG | (b) Transitive closure | (c) Measurements vector | (d) Root causes |

Figure 1: (a) A DAG for a river network. The weights capture fractions of pollution transported between adjacent nodes. (b) The transitive closure. The weights are fractions of pollution transported between all pairs of connected nodes. (c) A possible vector measuring pollution, and (d) the root causes of the pollution, sparse in this case.

**Data and root causes.** Fig. 1c shows a possible data vector $\mathbf{x}$ on the DAG, for example, the pollution measurement at each node done once a day (and without noise in this case). The measurement is the accumulated pollution from all upstream nodes. Within the model, the associated root causes $\mathbf{c}$ in Fig. 1d then show the origin of the pollution, two in this case. Sparsity in $\mathbf{C}$ means that each day only a small number of cities pollute. Negligible pollution from other sources is captured by noise $\mathbf{N}_c$ and $\mathbf{N}_x$ models the noise in the pollution measurements (both are assumed to be 0 in Fig. 1).

We view the pollution model as an abstraction that can be transferred to other real-world scenarios. For example, in gene networks that measure gene expression, few root causes would mean that few genes are activated in a considered dataset. In a citation network where one measures the impact of keywords/ideas, few root causes would correspond to the few origins of them.

## 3 Learning the DAG

In this section we present our approach for recovering the DAG adjacency matrix $\mathbf{A}$ from given data $\mathbf{X}$ under the assumption of few root causes, i.e., (4) (or (5)) and (6). We first show that under no measurement noise, our proposed setting is identifiable. Then we theoretically analyze our data generating model in the absence of noise and prove that the true DAG adjacency $\mathbf{A}$ is the global minimizer of the $L^0$-norm of the root causes. Finally, to learn the DAG in practice, we perform a continuous relaxation to obtain an optimization problem that is solvable with differentiation.

### 3.1 Identifiability

The rows of the data $\mathbf{X}$ generated by a linear SEM are commonly interpreted as observations of a random row vector $X = (X_1, ..., X_d)$ [Peters and Bühlmann, 2014, Zheng et al., 2018], i.e., it is written analogous to (2) as $X_i = X\mathbf{A}_{:,i} + N_i$, where $N_i$ are i.i.d. zero-mean random noise variables, or, equivalently, as $X = X\mathbf{A} + N$. Given a distribution $P_N$ of the noise vector, the DAG with graph adjacency matrix $\mathbf{A}$ is then called identifiable if it is uniquely determined by the distribution $P_X$ of $X$. Shimizu et al. [2006] state that $\mathbf{A}$ is identifiable if all $N_i$ are non-Gaussian noise variables. This applies directly to our setting and yields the following result.

**Theorem 3.1.** *Assume the data generation $X = (C + N_c)(\mathbf{I} + \overline{\mathbf{A}})$, where $C$ and $N_c$ are independent. Let $p \in [0, 1)$ and assume the $C_i$ are independent random variables taking uniform values from $[0, 1]$ with probability $p$, and are $= 0$ with probability $1 - p$. The noise vector $N_c$ is defined as before. Then the DAG given by $\mathbf{A}$ is identifiable.*

*Proof.* Using (5), the data generation equation can be viewed as a linear SEM (in standard recursive form) with noise variable $C + N_c$, which is non-Gaussian due to Lévy-Cramér decomposition theorem [Lévy, 1935, Cramér, 1936], because $C$ is non-Gaussian. The statement then follows from LiNGAM [Shimizu et al., 2006], given that the noise variables of $C$ and $N_c$ are independent. Note that this independency would be violated if we assumed non-zero measurement noise $N_x$, as in (4). $\square$

In Theorem 3.1, $C$ yields sparse observations whenever $p$ is close to zero (namely $dp$ root causes in expectation). However, identifiability follows for all $p$ due to non-Gaussianity.

## 3.2 $L^0$ minimization problem and global minimizer

Suppose that the data $\mathbf{X}$ are generated via the noise-free version of (4):

$$\mathbf{X} = \mathbf{C} \left(\mathbf{I} + \overline{\mathbf{A}}\right).\tag{7}$$

We assume $\mathbf{C}$ to be generated as in Theorem 3.1, i.e., with randomly uniform values from $[0, 1]$ with probability $p$ and 0 with probability $1 - p$, where $p$ is small such that $\mathbf{C}$ is sparse. Given the data $\mathbf{X}$ we aim to find $\mathbf{A}$ by enforcing maximal sparsity on the associated $\mathbf{C}$, i.e., by minimizing $\|\mathbf{C}\|_0$:

$$\min_{\mathbf{A} \in \mathbb{R}^{d \times d}} \left\| \mathbf{X} \left(\mathbf{I} + \overline{\mathbf{A}}\right)^{-1} \right\|_0 \quad \text{s.t.} \quad \mathbf{A} \text{ is acyclic.}\tag{8}$$

The following Theorem 3.2 states that given enough data, the true $\mathbf{A}$ is the unique global minimizer of (8). This means that in that case the optimization problem (8) correctly specifies the true DAG. One can view the result as a form of concrete, non-probabilistic identifiability. Note that Theorem 3.2 is not a sample-complexity result, but a statement of exact reconstruction in the absence of noise. The sample complexity of our algorithm is empirically evaluated later.

**Theorem 3.2.** *Consider a DAG with weighted adjacency matrix $\mathbf{A}$ with $d$ nodes. Given exponential (in $d$) number $n$ of samples $\mathbf{X}$ the matrix $\mathbf{A}$ is, with high probability, the global minimizer of the optimization problem* (8).

*Proof.* See Appendix C. $\qquad\square$

## 3.3 Continuous relaxation

In practice the optimization problem (8) is too expensive to solve due its combinatorial nature, and, of course, the noise-free assumption rarely holds in real-world data. We now consider again our general data generation in (4) assuming sparse root causes $\mathbf{C}$ and noises $\mathbf{N}_c, \mathbf{N}_x$ satisfying the criterion (6). To solve it we consider a continuous relaxation of the optimization objective. Typically (e.g., [Zheng et al., 2018, Lee et al., 2019, Bello et al., 2022]), continuous optimization DAG learning approaches have the following general formulation:

$$\begin{aligned} \min_{\mathbf{A} \in \mathbb{R}^{d \times d}} & \ell\left(\mathbf{X}, \mathbf{A}\right) + R\left(\mathbf{A}\right) \\ \text{s.t.} \quad & h\left(\mathbf{A}\right) = 0, \end{aligned}\tag{9}$$

where $\ell\left(\mathbf{A}, \mathbf{X}\right)$ is the loss function corresponding to matching the data, $R\left(\mathbf{A}\right)$ is a regularizer that promotes sparsity in the adjacency matrix, usually equal to $\lambda \|\mathbf{A}\|_1$, and $h\left(\mathbf{A}\right)$ is a continuous constraint enforcing acyclicity.

In our case, following a common practice in the literature, we substitute the $L^0$-norm from (8) with its convex relaxation [Ramirez et al., 2013], the $L^1$-norm. Doing so allows for some robustness to (low magnitude) noise $\mathbf{N}_c$ and $\mathbf{N}_x$. We capture the acyclicity with the continuous constraint $h\left(\mathbf{A}\right) = tr\left(e^{\mathbf{A} \odot \mathbf{A}}\right) - d$ used by Zheng et al. [2018], but could also use the form of Yu et al. [2019]. As sparsity regularizer for the adjacency matrix and we choose $R\left(\mathbf{A}\right) = \lambda \|\mathbf{A}\|_1$. Hence, our final continuous optimization problem is

$$\min_{\mathbf{A} \in \mathbb{R}^{d \times d}} \frac{1}{2n} \left\| \mathbf{X} \left(\mathbf{I} + \overline{\mathbf{A}}\right)^{-1} \right\|_1 + \lambda \|\mathbf{A}\|_1 \quad \text{s.t.} \quad h\left(\mathbf{A}\right) = 0.\tag{10}$$

We call this method SparseRC (sparse root causes).

## 4 Related work

The approaches to solving the DAG learning problem fall into two categories: combinatorial search or continuous relaxation. In general, DAG learning is an NP-hard problem since the combinatorial space of possible DAGs is super-exponential in the number of vertices [Chickering et al., 2004]. Thus, methods that search the space of possible DAGs apply combinatorial strategies to find an approximation of the ground truth DAG [Ramsey et al., 2017, Chickering, 2002, Tsamardinos et al., 2006].

**Continuous optimization.** Lately, with the advances of deep learning, researchers have been focusing on continuous optimization methods [Vowels et al., 2021] modelling the data generation process using SEMs. Among the first methods to utilize SEMs were CAM [Bühlmann et al., 2014] and LiNGAM [Shimizu et al., 2006], which specializes to linear SEMs with non-Gaussian noise. NOTEARS by Zheng et al. [2018] formulates the combinatorial constraint of acyclicity as a continuous one, which enables the use of standard optimization algorithms. Despite concerns, such as lack of scale-invariance [Kaiser and Sipos, 2022, Reisach et al., 2021], it has inspired many subsequent DAG learning methods. The current state-of-the-art of DAG learning methods for linear SEMs include DAGMA by Bello et al. [2022], which introduces a log-det acyclicity constraint, and GOLEM by Ng et al. [2020], which studies the role of the weighted adjacency matrix sparsity, the acyclicity constraint, and proposes to directly minimize the data likelihood. Zheng et al. [2020] extended NOTEARS to apply in non-linear SEMs. Other nonlinear methods for DAG learning include DAG-GNN by Yu et al. [2019], in which also a more efficient acyclicity constraint than the one in NOTEARS is proposed, and DAG-GAN by Gao et al. [2021]. DAG-NoCurl by Yu et al. [2021] proposes learning the DAG on the equivalent space of weighted gradients of graph potential functions. Pamfil et al. [2020] implemented DYNOTEARS, a variation of NOTEARS, compatible with time series data. A recent line of works considers permutation-based methods to parameterize the search space of the DAG [Charpentier et al., 2022, Zantedeschi et al., 2022]. The work in this paper considers DAG learning under the new assumption of few root causes. In Misiakos et al. [2024] we further build on this work by learning graphs from time series data.

**Fourier analysis.** Our work is also related to the recently proposed form of Fourier analysis on DAGs from Seifert et al. [2023, 2022], where equation (3) appears as a special case. The authors argue that (what we call) the root causes can be viewed as a form of spectrum of the DAG data. This means the assumption of few root causes is equivalent to Fourier-sparsity, a frequent assumption for common forms of Fourier transform [Hassanieh, 2018, Stobbe and Krause, 2012, Amrollahi et al., 2019]. Experiments in [Seifert et al., 2023, Misiakos et al., 2024] reconstruct data from samples under this assumption.

## 5 Experiments

We experimentally evaluate our DAG learning method SparseRC with both synthetically generated data with few root causes and real data from gene regulatory networks [Sachs et al., 2005]. We also mention that our method was among the winning solutions [Misiakos et al., 2023, Chevalley et al., 2023] of a competition of DAG learning methods to infer causal relationships between genes [Chevalley et al., 2022]. The evaluation was done on non-public data by the organizing institution and is thus not included here.

**Benchmarks.** We compare against prior DAG learning methods suitable for data generated by linear SEMs with additive noise. In particular, we consider the prior GOLEM [Ng et al., 2020], NOTEARS [Zheng et al., 2018], DAGMA [Bello et al., 2022], DirectLiNGAM [Shimizu et al., 2011], PC [Spirtes et al., 2000] and the greedy equivalence search (GES) [Chickering, 2002]. We also compared SparseRC against LiNGAM [Shimizu et al., 2006], causal additive models (CAM) [Bühlmann et al., 2014], DAG-NoCurl [Yu et al., 2021], fast greedy equivalence search (fGES) [Ramsey et al., 2017], the recent simple baseline sortnregress [Reisach et al., 2021] and max-min hill-climbing (MMHC) [Tsamardinos et al., 2006], but they where not competitive and thus we only include the results in Appendix B.

**Metrics.** To evaluate the found approximation $\hat{\mathbf{A}}$ of the true adjacency matrix $\mathbf{A}$, we use common performance metrics as in [Ng et al., 2020, Reisach et al., 2021]. In particular, the structural Hamming distance (SHD), which is the number of edge insertions, deletions, or reverses needed to convert $\hat{\mathbf{A}}$ to $\mathbf{A}$, and the structural intervention distance (SID), introduced by Peters and Bühlmann [2015], which is the number of falsely estimated intervention distributions. SHD gives a good estimate of a method's performance when it is close to $0$. In contrast, when it is high it can either be that the approximated DAG has no edges in common with the ground truth, but it can also be that the approximated DAG contains all the edges of the original one together with some false positives. Thus, in such cases, we also consider the true positive rate (TPR) of the discovered edges. We also report the total number of nonzero edges discovered for the real dataset [Sachs et al., 2005] and, for the synthetic data, provide results reporting the TPR, SID, and normalized mean square error (NMSE) metrics in Appendix B. For the methods that can successfully learn the underlying graph we further

Table 1: SHD metric (lower is better) for learning DAGs with 100 nodes and 400 edges. Each row is an experiment. The first row is the default, whose settings are in the blue column. In each other row, exactly one default parameter is changed (Change column). The last six columns correspond to prior algorithms. The best results are shown in bold. Entries with SHD $> 400$ are reported as *failure* or shown in green if the TPR is $> 0.8$.

| | Hyperparameter | Default | Change | Varsort. | SparseRC (ours) | GOLEM | NOTEARS | DAGMA | DirectLiNGAM | PC | GES |
|---|---|---|---|---|---|---|---|---|---|---|---|
| 1 | Default settings | | | 0.95 | **0.6 ± 0.8** | 82 ± 34 | 59 ± 22 | 269 ± 6.0 | 282 ± 34 | 247 ± 9.0 | failure |
| 2 | Graph type | Erdös-R. | Scale-free | 0.99 | **2.2 ± 1.5** | 34 ± 9.0 | 28 ± 9.5 | 296 ± 14 | 188 ± 27 | 275 ± 16 | 375 ± 141 |
| 3 | $N_c, N_x$ distribution | Gaussian | Gumbel | 0.97 | **1.4 ± 1.0** | 87 ± 44 | 59 ± 17 | 278 ± 7.4 | 287 ± 43 | 250 ± 17 | 396 ± 33 |
| 4 | Edges / Vertices | 4 | 10 | 0.99 | **46 ± 7.5** | 212 ± 70 | 243 ± 26 | 896 ± 30 | 1078 ± 105 | 973 ± 17 | failure |
| 5 | Standardization | No | Yes | 0.50 | 624 ± 48 | failure | failure | failure | 278 ± 25 | **247 ± 18** | failure |
| 6 | Larger weights in $\mathbf{A}$ | (0.1, 0.9) | (0.5, 2) | 1.00 | failure | 96 ± 25 | **92 ± 14** | 209 ± 4.1 | 840 ± 121 | 400 ± 9.8 | failure |
| 7 | $N_c, N_x$ deviation | $\sigma = 0.01$ | $\sigma = 0.1$ | 0.97 | 504 ± 19 | **98 ± 14** | 199 ± 12 | 238 ± 13 | 538 ± 45 | 255 ± 11 | failure |
| 8 | Dense root causes $\mathbf{C}$ | $p = 0.1$ | $p = 0.5$ | 0.98 | 1221 ± 33 | **29 ± 2.5** | 126 ± 32 | 83 ± 8.0 | 257 ± 14 | 244 ± 12 | failure |
| 9 | Samples | $n = 1000$ | $n = 100$ | 0.97 | 2063 ± 92 | failure | failure | **328 ± 13** | error | 351 ± 12 | failure |
| 10 | Fixed support | No | Yes | 0.89 | failure | failure | failure | failure | failure | **379 ± 37** | failure |

proceed on approximating the true root causes $\mathbf{C}$ that generated the data $\mathbf{X}$ via (4). With regard to this aspect, we count the number of correctly detected root causes $\mathbf{C}$ TPR, their false positive rate $\mathbf{C}$ FPR and the weighted approximation $\mathbf{C}$ NMSE. We also report the runtime for all methods in seconds. For each performance metric, we compute the average and standard deviation over five repetitions of the same experiment.

**Our implementation.** To solve (10) in practice, we implemented[1] a PyTorch model with a trainable parameter representing the weighted adjacency matrix $\mathbf{A}$. This allows us to utilize GPUs for the acceleration of the execution of our algorithm (GOLEM uses GPUs, NOTEARS does not). Then we use the standard Adam [Kingma and Ba, 2014] optimizer to minimize the loss defined in (10).

### 5.1 Evaluation on data with few root causes

**Data generating process and defaults.** In the second, blue column of Table 1 we report the default settings for our experiment. We generate a random Erdös-Renyi graph with $d = 100$ nodes and assign edge directions to make it a DAG as in [Zheng et al., 2018]. The ratio of edges to vertices is set to $4$, so the number of edges is $400$. The entries of the weighted adjacency matrix are sampled uniformly at random from $(-b, -a) \cup (a, b)$, where $a = 0.1$ and $b = 0.9$. As in [Zheng et al., 2018, Ng et al., 2020, Bello et al., 2022] the resulting adjacency matrix is post-processed with thresholding. In particular, the edges with absolute weight less than the threshold $\omega = 0.09$ are discarded. Next, the root causes $\mathbf{C}$ are instantiated by setting each entry either to some random uniform value from $(0, 1)$ with probability $p = 0.1$ or to 0 with probability $1 - p = 0.9$ (thus, as in Theorem 3.1, the location of the root causes will vary). The data matrix $\mathbf{X}$ is computed according to (4), using Gaussian noise $N_c, N_x$ of standard deviation $\sigma = 0.01$ to enforce (6). Finally, we do not standardize (scale for variance $= 1$) the data, and $\mathbf{X}$ contains $n = 1000$ samples (number of rows).

**Experiment 1: Different application scenarios.** Table 1 compares SparseRC to six prior algorithms using the SHD metric. Every row corresponds to a different experiment that alters one particular hyperparameter of the default setting, which is the first row with values of the blue column as explained above. For example, the second row only changes the graph type from Erdös-Renyi to scale-free, while keeping all other settings. Note that in the last row, fixed support means that the location of the root causes is fixed for every sample in the data.

Since the graphs have 400 edges, an SHD $\ll 400$ can be considered as good and beyond 400 can be considered a failure. In row 3 this threshold is analogously set to 1000. The failure cases are indicated as such in Table 1, or the SHD is shown in green if they still achieve a good TPR $> 0.8$. In Appendix B.1 we report the TPR, SID, and the runtimes of all methods.

In the first three rows, we examine scenarios that perfectly match the condition (6) of few root causes. For the default settings (row 1), scale-free graphs (row 2), and different noise distribution (row 3) SparseRC performs best and almost perfectly detects all edges. The next experiments alter parameters that deteriorate the few root causes assumption. Row 4 considers DAGs with average degree 10, i.e., about 1000 edges. High degree can amplify the measurement noise and hinder the root causes assumption (6). However, our method still performs best, but even larger degrees decay the performance of all methods as shown in experiment 5 below. The standardization of data (row

---

[1]Our code is publicly available at https://github.com/pmisiakos/SparseRC.

Table 2: Runtime [seconds] report of the top-performing methods in Table 1.

| | Hyperparameter | Default | Change | SparseRC (ours) | GOLEM | NOTEARS |
|---|---|---|---|---|---|---|
| 1. | Default settings | | | $10 \pm 1.8$ | $529 \pm 210$ | $796 \pm 185$ |
| 2. | Graph type | Erdös-Renyi | Scale-free | $11 \pm 1.1$ | $460 \pm 184$ | $180 \pm 7.2$ |
| 3. | $\mathbf{N}_c, \mathbf{N}_x$ distribution | Gaussian | Gumbel | $8.2 \pm 0.7$ | $349 \pm 125$ | $251 \pm 48$ |
| 4. | Edges / Vertices | 4 | 10 | $14 \pm 1.0$ | $347 \pm 121$ | $471 \pm 82$ |
| 5. | Samples | $n = 1000$ | $n = 100$ | $13 \pm 0.7$ | $194 \pm 9.6$ | $679 \pm 72$ |
| 6. | Standardization | No | Yes | $11 \pm 1.9$ | $326 \pm 145$ | $781 \pm 76$ |
| 7. | Larger weights in $\mathbf{A}$ | $(0.1, 0.9)$ | $(0.5, 2)$ | $8.4 \pm 0.6$ | $431 \pm 177$ | $2834 \pm 228$ |
| 8. | $\mathbf{N}_c, \mathbf{N}_x$ deviation | $\sigma = 0.01$ | $\sigma = 0.1$ | $8.7 \pm 0.7$ | $309 \pm 63$ | $433 \pm 53$ |
| 9. | Dense root causes $\mathbf{C}$ | $p = 0.1$ | $p = 0.5$ | $9.1 \pm 0.7$ | $334 \pm 121$ | $427 \pm 35$ |
| 10. | Fixed support | No | Yes | $15 \pm 2.0$ | $360 \pm 142$ | $669 \pm 386$ |

5) is generally known to negatively affect algorithms with continuous objectives [Reisach et al., 2021] as is the case here. Also, standardization changes the relative scale of the root causes and as a consequence affects (6). Row 6 considers edge weights $> 1$ which amplifies the measurement noise in (6) and our method fails. Higher standard deviation in the root causes noise (row 7) or explicitly higher density in $\mathbf{C}$ (row 8) decreases performance overall. However, In both cases SparseRC still discovers a significant amount of the unknown edges. For a small number of samples (row 9) most methods fail. Ours achieves a high TPR but requires more data to converge to the solution as we will see in experiment 2 below. Row 10 is out of the scope of our method and practically all methods fail.

Overall, the performance of SparseRC depends heavily on the degree to which the assumption of few root causes (6) is fulfilled. The parameter choices in the table cover a representative set of possibilities from almost perfect recovery (row 1) to complete failure (row 10).

SparseRC is also significantly faster than the best competitors GOLEM and NOTEARS with typical speedups in the range of $10$–$50\times$ as shown in Table 2. It is worth mentioning that even though LiNGAM provides the identifiability Theorem 3.1 of our proposed setting, it is not able to recover the true DAG. While both DirectLiNGAM and LiNGAM come with theoretical guarantees for their convergence, these require conditions, such as infinite amount of data, which in practice are not met. We include a more extensive reasoning for their subpar performance in Appendix B.4 together a particularly designed experiment for LiNGAM and DirectLiNGAM.

**Varsortability.** In Table 1 we also include the varsortability for each experimental setting. Our measurements for Erdös-Renyi graphs (all rows except row 2) are typically $1$–$2\%$ lower than those reported in [Reisach et al., 2021, Appendix G.1] for linear SEMs, but still high in general. However, the trivial variance-sorting method sortnregress, included in Appendix B, fails overall. Note again that for fixed sparsity support (last row), all methods fail and varsortability is lower. Therefore, in this scenario, our data generating process poses a hard problem for DAG learning.

**Experiment 2: Varying number of nodes or samples.** In this experiment we first benchmark SparseRC with varying number of nodes (and thus number of edges) in the ground truth DAG, while keeping the data samples in $\mathbf{X}$ equal to $n = 1000$. Second, we vary the number of samples, while keeping the number of nodes fixed to $d = 100$. All other parameters are set to default. The results are shown in Fig. 2 reporting SHD, SID, and the runtime.

When varying the number of nodes (first row in Fig. 2) SparseRC, GOLEM, and NOTEARS achieve very good performance whereas the other methods perform significantly worse. As the number of nodes increases, SparseRC performs best both in terms of SHD and SID while being significantly faster than GOLEM and NOTEARS. When increasing the number of samples (second row) the overall performances improve. For low number of samples SparseRC, GOLEM and NOTEARS fail. Their performance significantly improves after 500 samples where SparseRC overall achieves the best result. The rest of the methods have worse performance. SparseRC is again significantly faster than GOLEM and NOTEARS in this case.

**Experiment 3: Learning the root causes.** Where our method succeeds we can also recover the root causes that generate the data. Namely, if we recover a very good estimate of the true adjacency matrix via (10), we may compute an approximation $\widehat{\mathbf{C}}$ of the root causes $\mathbf{C}$, up to noise, by solving (4):

$$\widehat{\mathbf{C}} = \mathbf{C} + \mathbf{N}_c + \mathbf{N}_x \left(\mathbf{I} - \mathbf{A}\right) = \mathbf{X} \left(\mathbf{I} + \overline{\mathbf{A}}\right)^{-1}. \tag{11}$$

In the last row of Fig. 2 we evaluate the top-performing methods on the recovery of the root causes (and the associated values) with respect to detecting their locations ($\mathbf{C}$ TPR and FPR) and recovering

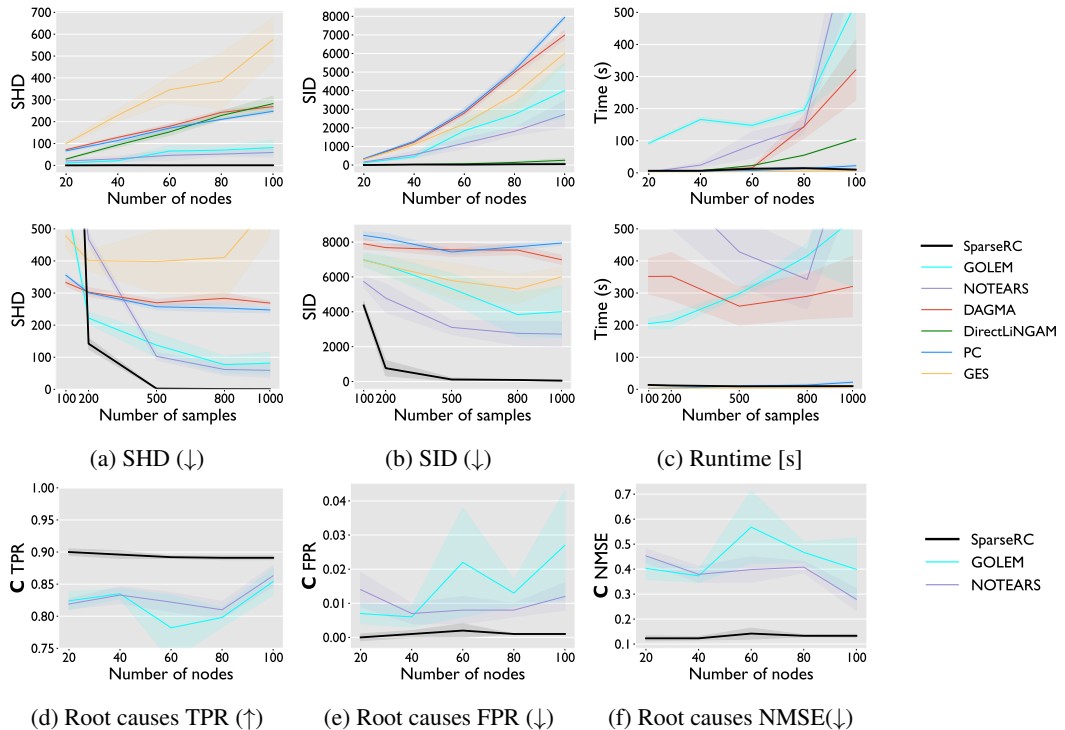

Figure 2: Performance report on the default settings. (a,b,c) illustrate SHD, SID and runtime (lower is better) while varying the number of nodes with 1000 samples (first row) or varying the number of samples with 100 nodes (second row). (d, e) illustrate TPR and FPR of the estimated support of $\mathbf{C}$, and (f) reports the accuracy of estimating $\mathbf{C}$ as NMSE.

Table 3: Performance of the top-performing methods on larger DAGs.

| Nodes $d$, samples $n$ | SparseRC | NOTEARS | GOLEM | SparseRC | NOTEARS | GOLEM |
|---|---|---|---|---|---|---|
| $d = 200$, $n = 500$ | 22 | 155 | 281 | 21 | 1061 | 600 |
| $d = 500$, $n = 1000$ | 27 | 245 | 574 | 204 | 5789 | 5428 |
| $d = 1000$, $n = 5000$ | 26 | 282 | 699 | 1085 | 10104 | 37290 |
| $d = 2000$, $n = 10000$ | 50 | 489 | time-out | 7141 | 46213 | time-out |
| $d = 3000$, $n = 10000$ | 134 | time-out | time-out | 19660 | time-out | time-out |

(a) SHD ($\downarrow$)

(b) Runtime (s)

their numerical values using the $\mathbf{C}$ NMSE metric. The support is chosen as the entries $(i, j)$ of $\mathbf{C}$ that are in magnitude within $10\%$ of the largest value in $\mathbf{C}$ (i.e., relatively large): $c_{ij} > 0.1 \cdot \mathbf{C}_{\max}$. We consider the default experimental settings.

**Experiment 4: Larger DAGs.** In Table 3, we investigate the scalability of the best methods SparseRC, GOLEM, and NOTEARS from the previous experiments. We consider five different settings up to $d = 3000$ nodes and $n = 10000$ samples. We use the prior default settings except for the sparsity of the root causes, which is now set to $5\%$. The metrics here are SHD and the runtime . The results show that SparseRC excels in the very sparse setting with almost perfect reconstruction, far outperforming the others including in runtime. SparseRC even recovers the edge weights with an average absolute error of about $5\%$ for each case in Table 3, as shown in Appendix B.3.

**Experiment 5: Varying average degree.** We evaluate the top-scoring methods on DAGs with higher average degree (dense DAGs). Note that this violates the typical assumption of sparse DAGs [Zheng et al., 2018]. In Fig. 3 we consider default settings with DAGs of average degree up to 20. SID was not computed due to the presence of cycles in the output of the algorithms and TPR is reported instead. We conclude that for dense DAGs the performance decays, as expected.

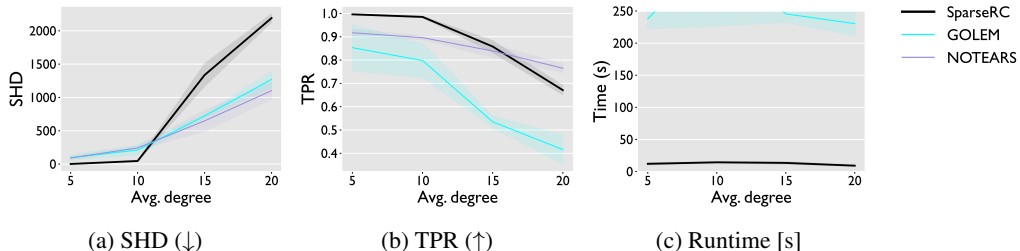

(a) SHD (↓)          (b) TPR (↑)          (c) Runtime [s]

Figure 3: Evaluation of the top-performing methods on denser DAGs.

Table 4: Performance of the top-performing methods on the dataset by Sachs et al. [2005].

|          | SHD ↓ | SID ↓ | Total edges |
|----------|-------|-------|-------------|
| SparseRC | 15    | 45    | 16          |
| NOTEARS  | **11**| 44    | 15          |
| GOLEM    | 21    | **43**| 19          |
| DAGMA    | 14    | 46    | 11          |

## 5.2 Evaluation on a real dataset

We apply SparseRC on the causal protein-signaling network data provided by Sachs et al. [2005]. The dataset consists of $7466$ samples (we use the first $853$) for a network with $11$ nodes that represent proteins, and $17$ edges representing their interactions. It is small, but the task of learning it has been considered a difficult benchmark in [Ng et al., 2020, Gao et al., 2021, Yu et al., 2019, Zheng et al., 2018]. It is not known whether the assumption of few root causes holds in this case. We report the performance metrics SHD and SID for the most successful methods in Table 4. The number of total edges is used to ensure that the output of the methods are non-trivial (e.g. empty graph).

The best SID is achieved by GOLEM (equal variance), which, however, has higher SHD. NOTEARS has the best SHD equal to $11$. Overall, SparseRC performs well, achieving the closest number of edges to the real one with $16$ and a competitive SHD and SID. We also mention again the good performance of SparseRC on a non-public biological dataset [Misiakos et al., 2023, Chevalley et al., 2023] as part of the competition by Chevalley et al. [2022].

## 6 Broader Impact and Limitations

Our method inherits the broader impact of prior DAG learning methods including the important caveat for practical use that the learned DAG may not represent causal relations, whose discovery requires interventions. Further limitations that we share with prior work include (a) Learning DAGs beyond 10000 nodes are out of reach, (b) there is no theoretical convergence guarantee for the case that includes noise, (c) empirically, the performance drops in low varsortability, (d) our method is designed for linear SEMs like most of the considered benchmarks.

A specific limitation of our contribution is that it works well only for few root causes of varying location in the dataset. This in addition implies that the noise must have low deviation and the graph to have low average degree and weights of magnitude less than one. Also, in our experiments, we only consider root causes with support that follows a multivariate Bernoulli distribution.

## 7 Conclusion

We presented a new perspective on linear SEMs by introducing the notion of root causes. Mathematically, this perspective translates (or solves) the recurrence describing the SEM into an invertible linear transformation that takes as input DAG data, which we call root causes, to produce the observed data as output. Prior data generation for linear SEMs assumed a dense, random valued input vector. In this paper, we motivated and studied the novel scenario of data generation and DAG learning for few root causes, i.e., a sparse input with noise, and noise in the measurement data. Our solution in this setting performs significantly better than prior algorithms, in particular for high sparsity where it can even recover the edge weights, is scalable to thousands of nodes, and thus expands the set of DAGs that can be learned in real-world scenarios where current methods fail.

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

## A  Few Root Causes Assumption

**Lemma A.1.** *Consider random row vector $C \in \mathbb{R}^{1 \times d}$ of root causes that are generated as in Section 5.1 with probability $p = 0.1$ being non-zero and value taken uniformly at random from $(0, 1)$. Also, let the noise vectors $N_c, N_x \in \mathbb{R}^{1 \times d}$ be Gaussian $N_c, N_x \sim \mathcal{N}(\mathbf{0}, \sigma\mathbf{I})$ with $\sigma = 0.01$ and the DAG has average degree $\delta(G) = 4$ and weights $a_{ij} \in [-1, 1]$. Then we show that for $\epsilon = \delta = 0.1$:*

$$\frac{\mathbb{E}\left[\|C\|_0\right]}{d} \leq \epsilon \quad \text{(few root causes)},$$

$$\frac{\mathbb{E}\left[\|N_c + N_x\left(\mathbf{I} - \mathbf{A}\right)\|_1\right]}{\mathbb{E}\left[\|C\|_1\right]} \leq \delta \quad \text{(negligible noise)}, \tag{12}$$

*Proof.* From the Bernoulli distribution for the support of the root causes we get

$$\mathbb{E}\left[\|C\|_0\right] = \frac{pd}{d} = 0.1 \tag{13}$$

Also the root causes take uniform values in $(0, 1)$ and have expected value $= 0.5$. Therefore,

$$\mathbb{E}\left[\|C\|_1\right] = 0.5d \tag{14}$$

Finally, we have that:

$$\mathbb{E}\left[\|N_c + N_x\left(\mathbf{I} - \mathbf{A}\right)\|_1\right] \leq$$
$$\mathbb{E}\left[\|N_c\|_1\right] + \mathbb{E}\left[\|N_x\left(\mathbf{I} - \mathbf{A}\right)\|_1\right] \leq \tag{15}$$
$$\mathbb{E}\left[\|N_c\|_1\right] + (\delta(G) + 1)\mathbb{E}\left[\|N_x\|_1\right] =$$

$$6d\sigma\sqrt{\frac{2}{\pi}} \tag{16}$$

The computation of the expected value follows from the mean of the Folded normal distribution [Tsagris et al., 2014]. Dividing the last two relations gives the required relation

$$\frac{\mathbb{E}\left[\|N_c + N_x\left(\mathbf{I} - \mathbf{A}\right)\|_1\right]}{\mathbb{E}\left[\|C\|_1\right]} \leq 0.1 \cdot \frac{6}{5}\sqrt{\frac{2}{\pi}} < \delta \tag{17}$$

$\square$

## B  Additional experimental results

### B.1  Different application scenarios

**Experiment 1: Different application scenarios.** We present additional experimental results to include more metrics and baselines. The results support our observations and conclusions from the main text. First, we expand Table 1 reporting the SHD metric and provide the computation of SHD, TPR, SID and runtime in the tables below. Those further include the methods LiNGAM [Shimizu et al., 2006], CAM [Bühlmann et al., 2014], DAG-NoCurl [Yu et al., 2021], fGES [Ramsey et al., 2017], sortnregress [Reisach et al., 2021] and MMHC [Tsamardinos et al., 2006] .

## B.2 Varying number of nodes or samples

**Experiment 2: Varying number of nodes or samples.** In addition to Fig. 2 we include the plots of Fig. 4 for the experiments that vary the number of nodes of the ground truth DAG or the number of samples in the data. The plots include the methods LiNGAM, CAM, DAG-NoCurl, fGES, sortnregress and MMHC and additionally contain the metrics TPR and total number of edges regarding the unweighted adjacency matrix and NMSE with respect to the weighted approximation of the adjacency matrix.

## B.3 Larger DAGs

**Experiment 3: Larger DAGs.** In the large scale experiment we don't report SID as it is computationally too expensive for a large number of nodes. Since, our method achieved perfect reconstruction, we further investigate whether the true weights were recovered. Table 9 reports metrics that evaluate the weighted approximation of the true adjacency matrix. Denoting with $|E|$ the number of edges of the ground truth DAG $\mathcal{G} = (V, E)$, we compute:

- the average $L^1$ loss $\frac{\|\mathbf{A} - \widehat{\mathbf{A}}\|_1}{|E|}$,

- the Max-$L^1$ loss $\max_{i,j} \left| \mathbf{A}_{ij} - \widehat{\mathbf{A}}_{ij} \right|$,

- the average $L^2$ loss $\frac{\|\mathbf{A} - \widehat{\mathbf{A}}\|_2}{|E|}$ and

- the NMSE $\frac{\|\mathbf{A} - \widehat{\mathbf{A}}\|_2}{\|\mathbf{A}\|_2}$.

## B.4 LiNGAM's performance

The subpar performance of DirectLiNGAM arises as a contradiction to the fact that [Shimizu et al., 2006] provide the identifiability result for our setting. A possible reasoning to this can be that the corresponding linear noise variances as in (5) are non i.i.d., where as both LiNGAM and DirectLiNGAM consider i.i.d. noise distribution. In Fig. 5, we evaluate LiNGAM and DirectLiNGAM with respect to ours, using default settings, but with zero measurement noise $\mathbf{N}_x = 0$, which translates (4) to a linear SEM with i.i.d. noise. We conclude that the methods cannot recover the true graph even in these settings. The reasoning we give for their failure is the following: First, LiNGAM can stay on a local optimum due to a badly chosen initial state in the ICA step of the algorithm, or even compute a wrong ordering of the variables [Shimizu et al., 2011]. Secondly, DirectLiNGAM is guaranteed to converge to the true solution only if the conditions are striclty met, among which is the infinite number of available data, which is not the case in our experiments.

## B.5 Real Data

In the real dataset from [Sachs et al., 2005] we follow Lachapelle et al. [2019] and only use the first 853 samples from the dataset. Differences may occur between the reported results and the literature due to different choice of hyperparameters and the use of 853 samples, where others might utilize the full dataset.

**Root causes in real data.** To question whether the few root causes assumption is valid in the real dataset of [Sachs et al., 2005] we perform the following experiment. Using the ground truth matrix in the [Sachs et al.,

Table 5: SHD metric (lower is better) for learning DAGs with 100 nodes and 400 edges. Each row is an experiment. The first row is the default, whose settings are in the blue column. In each other row, exactly one default parameter is changed (change). The best results are shown in bold. Entries with SHD higher that $400$ are reported as *failure* and in such a case the corresponding TPR is reported if it is higher than $0.8$.

|  | Hyperparameter | Default | Change | LiNGAM | CAM | DAG-NoCurl | fGES | sortnregress | MMHC |
|---|---|---|---|---|---|---|---|---|---|
| 1. | Default settings | | | $\mathbf{285 \pm 38}$ | $325 \pm 28$ | $737 \pm 94$ | $547 \pm 86$ | $800 \pm 136$ | failure |
| 2. | Graph type | Erdös-Renyi | Scale-free | $\mathbf{187 \pm 28}$ | failure | $247 \pm 45$ | $382 \pm 64$ | $300 \pm 36$ | failure |
| 3. | $\mathbf{N}_c, \mathbf{N}_x$ distribution | Gaussian | Gumbel | $478 \pm 342$ | $\mathbf{350 \pm 18}$ | $642 \pm 55$ | $497 \pm 38$ | $796 \pm 86$ | failure |
| 4. | Edges / Vertices | 4 | 10 | $1102 \pm 108$ | $\mathbf{961 \pm 20}$ | $1455 \pm 277$ | failure | $1950 \pm 97$ | failure |
| 5. | Samples | $n = 1000$ | $n = 100$ | error | time-out | failure | error | error | failure |
| 6. | Standardization | No | Yes | $365 \pm 38$ | $\mathbf{329 \pm 34}$ | failure | $570 \pm 109$ | failure | failure |
| 7. | Larger weights in $\mathbf{A}$ | $(0.1, 0.9)$ | $(0.5, 2)$ | $860 \pm 129$ | failure | $\mathbf{135 \pm 56}$ | failure | $1275 \pm 133$ | failure |
| 8. | $\mathbf{N}_c, \mathbf{N}_x$ deviation | $\sigma = 0.01$ | $\sigma = 0.1$ | failure | $\mathbf{341 \pm 29}$ | failure | failure | $906 \pm 53$ | failure |
| 9. | Dense root causes $\mathbf{C}$ | $p = 0.1$ | $p = 0.5$ | failure | $\mathbf{271 \pm 21}$ | $586 \pm 58$ | $528 \pm 72$ | $746 \pm 134$ | failure |
| 10. | Fixed support | No | Yes | failure | $\mathbf{390 \pm 51}$ | failure | failure | failure | failure |

Table 6: TPR (higher is better). Entries with TPR lower that $0.5$ are reported as *failure*.

(a)

| | Hyperparameter | Default | Change | SparseRC (ours) | GOLEM | NOTEARS | DAGMA | DirectLiNGAM | PC | GES |
|---|---|---|---|---|---|---|---|---|---|---|
| 1. | Default settings | | | $\mathbf{1.00 \pm 0.00}$ | $0.82 \pm 0.09$ | $0.92 \pm 0.02$ | failure | $0.99 \pm 0.00$ | failure | $0.72 \pm 0.04$ |
| 2. | Graph type | Erdös-Renyi | Scale-free | $0.99 \pm 0.00$ | $0.95 \pm 0.01$ | $0.95 \pm 0.02$ | failure | $0.99 \pm 0.01$ | failure | $0.84 \pm 0.09$ |
| 3. | $\mathbf{N}_c, \mathbf{N}_x$ distribution | Gaussian | Gumbel | $\mathbf{1.00 \pm 0.00}$ | $0.82 \pm 0.11$ | $0.92 \pm 0.01$ | failure | $0.99 \pm 0.00$ | failure | $0.81 \pm 0.02$ |
| 4. | Edges / Vertices | 4 | 10 | $0.98 \pm 0.00$ | $0.80 \pm 0.07$ | $0.90 \pm 0.01$ | failure | $\mathbf{0.99 \pm 0.00}$ | failure | failure |
| 5. | Samples | $n = 1000$ | $n = 100$ | $0.90 \pm 0.02$ | failure | $0.75 \pm 0.02$ | failure | error | failure | $0.61 \pm 0.05$ |
| 6. | Standardization | No | Yes | $0.84 \pm 0.01$ | failure | failure | failure | $\mathbf{0.99 \pm 0.01}$ | failure | $0.78 \pm 0.09$ |
| 7. | Larger weights in $\mathbf{A}$ | $(0.1, 0.9)$ | $(0.5, 2)$ | failure | $0.85 \pm 0.04$ | $0.89 \pm 0.01$ | $0.53 \pm 0.01$ | $\mathbf{1.00 \pm 0.00}$ | failure | $0.70 \pm 0.09$ |
| 8. | $\mathbf{N}_c, \mathbf{N}_x$ deviation | $\sigma = 0.01$ | $\sigma = 0.1$ | $0.88 \pm 0.01$ | $0.85 \pm 0.03$ | $0.86 \pm 0.01$ | failure | $\mathbf{0.97 \pm 0.01}$ | failure | $0.70 \pm 0.04$ |
| 9. | Dense root causes $\mathbf{C}$ | $p = 0.1$ | $p = 0.5$ | $0.88 \pm 0.02$ | $0.94 \pm 0.01$ | $0.93 \pm 0.00$ | $0.81 \pm 0.02$ | $\mathbf{0.99 \pm 0.00}$ | failure | $0.80 \pm 0.05$ |
| 10. | Fixed support | No | Yes | failure | failure | failure | failure | $0.57 \pm 0.02$ | failure | $\mathbf{0.64 \pm 0.05}$ |

(b)

| | Hyperparameter | Default | Change | LiNGAM | CAM | DAG-NoCurl | fGES | sortnregress | MMHC |
|---|---|---|---|---|---|---|---|---|---|
| 1. | Default settings | | | $0.99 \pm 0.00$ | failure | $0.84 \pm 0.02$ | $0.80 \pm 0.04$ | $0.87 \pm 0.02$ | $0.58 \pm 0.03$ |
| 2. | Graph type | Erdös-Renyi | Scale-free | $\mathbf{0.99 \pm 0.01}$ | failure | $0.92 \pm 0.02$ | $0.76 \pm 0.06$ | $0.96 \pm 0.01$ | failure |
| 3. | $\mathbf{N}_c, \mathbf{N}_x$ distribution | Gaussian | Gumbel | $0.94 \pm 0.10$ | failure | $0.82 \pm 0.02$ | $0.82 \pm 0.03$ | $0.88 \pm 0.02$ | $0.56 \pm 0.04$ |
| 4. | Edges / Vertices | 4 | 10 | $0.98 \pm 0.01$ | failure | $0.84 \pm 0.05$ | $0.54 \pm 0.04$ | $0.90 \pm 0.01$ | failure |
| 5. | Samples | $n = 1000$ | $n = 100$ | error | time-out | $0.76 \pm 0.03$ | error | error | failure |
| 6. | Standardization | No | Yes | $0.94 \pm 0.01$ | failure | failure | $0.80 \pm 0.03$ | failure | $0.57 \pm 0.05$ |
| 7. | Larger weights in $\mathbf{A}$ | $(0.1, 0.9)$ | $(0.5, 2)$ | $\mathbf{1.00 \pm 0.00}$ | failure | $0.92 \pm 0.04$ | $0.74 \pm 0.06$ | $0.94 \pm 0.01$ | failure |
| 8. | $\mathbf{N}_c, \mathbf{N}_x$ deviation | $\sigma = 0.01$ | $\sigma = 0.1$ | $0.64 \pm 0.12$ | failure | $0.78 \pm 0.01$ | $0.76 \pm 0.04$ | $0.85 \pm 0.01$ | $0.58 \pm 0.02$ |
| 9. | Dense root causes $\mathbf{C}$ | $p = 0.1$ | $p = 0.5$ | $0.70 \pm 0.03$ | $0.51 \pm 0.03$ | $0.84 \pm 0.04$ | $0.83 \pm 0.03$ | $0.88 \pm 0.02$ | $0.57 \pm 0.04$ |
| 10. | Fixed support | No | Yes | $0.52 \pm 0.04$ | failure | failure | $0.54 \pm 0.03$ | $0.58 \pm 0.04$ | failure |

Table 7: SID (lower is better).

(a)

| | Hyperparameter | Default | Change | SparseRC (ours) | GOLEM | NOTEARS | DAGMA | DirectLiNGAM | PC | GES |
|---|---|---|---|---|---|---|---|---|---|---|
| 1. | Default settings | | | $\mathbf{51 \pm 66}$ | $4005 \pm 1484$ | $2720 \pm 720$ | $6989 \pm 268$ | $257 \pm 94$ | $7944 \pm 118$ | $6006 \pm 561$ |
| 2. | Graph type | Erdös-Renyi | Scale-free | $\mathbf{72 \pm 42}$ | $442 \pm 71$ | $425 \pm 59$ | $1581 \pm 172$ | $98 \pm 64$ | $2570 \pm 314$ | $1032 \pm 371$ |
| 3. | $\mathbf{N}_c, \mathbf{N}_x$ distribution | Gaussian | Gumbel | $\mathbf{125 \pm 91}$ | $4208 \pm 1767$ | $2945 \pm 601$ | $7697 \pm 220$ | $214 \pm 133$ | $7986 \pm 406$ | $4957 \pm 577$ |
| 4. | Edges / Vertices | 4 | 10 | $1190 \pm 149$ | $7174 \pm 1083$ | $5914$ | $8833 \pm 95$ | $\mathbf{808 \pm 156}$ | SID time-out | SID time-out |
| 5. | Samples | $n = 1000$ | $n = 100$ | SID time-out | $7410 \pm 326$ | SID time-out | SID time-out | error | SID time-out | SID time-out |
| 6. | Standardization | No | Yes | SID time-out | $9381 \pm 147$ | $8937 \pm 30$ | $8851 \pm 185$ | $\mathbf{293 \pm 174}$ | $7801 \pm 358$ | $5268 \pm 1078$ |
| 7. | Larger weights in $\mathbf{A}$ | $(0.1, 0.9)$ | $(0.5, 2)$ | $6988 \pm 305$ | $3165 \pm 549$ | $2379 \pm 292$ | $6624 \pm 351$ | $17 \pm 34$ | SID time-out | SID time-out |
| 8. | $\mathbf{N}_c, \mathbf{N}_x$ deviation | $\sigma = 0.01$ | $\sigma = 0.1$ | $7140 \pm 411$ | $3898 \pm 635$ | $4398 \pm 238$ | $7227 \pm 373$ | $882 \pm 157$ | $7841 \pm 306$ | $6810 \pm 792$ |
| 9. | Dense root causes $\mathbf{C}$ | $p = 0.1$ | $p = 0.5$ | SID time-out | $1871 \pm 256$ | $3040 \pm 257$ | $4539 \pm 383$ | $\mathbf{275 \pm 102}$ | SID time-out | SID time-out |
| 10. | Fixed support | No | Yes | $7605 \pm 259$ | $8110 \pm 413$ | $7707 \pm 275$ | $7749 \pm 310$ | $6800 \pm 601$ | $8119 \pm 314$ | $5817 \pm 759$ |

(b)

| | Hyperparameter | Default | Change | LiNGAM | CAM | DAG-NoCurl | fGES | sortnregress | MMHC |
|---|---|---|---|---|---|---|---|---|---|
| 1. | Default settings | | | $595 \pm 212$ | $8494 \pm 476$ | $5320 \pm 594$ | $7706 \pm 456$ | $4418 \pm 464$ | SID time-out |
| 2. | Graph type | Erdös-Renyi | Scale-free | $99 \pm 45$ | SID time-out | $807 \pm 219$ | $7412 \pm 493$ | $427 \pm 83$ | SID time-out |
| 3. | $\mathbf{N}_c, \mathbf{N}_x$ distribution | Gaussian | Gumbel | $1796 \pm 2754$ | $8916 \pm 317$ | $5723 \pm 575$ | $9040 \pm 170$ | $4522 \pm 797$ | SID time-out |
| 4. | Edges / Vertices | 4 | 10 | $1516 \pm 234$ | $9540 \pm 30$ | $6200 \pm 620$ | $9057 \pm 172$ | $4967 \pm 557$ | SID time-out |
| 5. | Samples | $n = 1000$ | $n = 100$ | error | time-out | $\mathbf{4496 \pm 507}$ | error | error | SID time-out |
| 6. | Standardization | No | Yes | $2847 \pm 577$ | $8707 \pm 287$ | $8955 \pm 68$ | $7872 \pm 605$ | $7945 \pm 124$ | SID time-out |
| 7. | Larger weights in $\mathbf{A}$ | $(0.1, 0.9)$ | $(0.5, 2)$ | $\mathbf{16 \pm 33}$ | $8147 \pm 234$ | $1660 \pm 781$ | $7137 \pm 496$ | $958 \pm 181$ | SID time-out |
| 8. | $\mathbf{N}_c, \mathbf{N}_x$ deviation | $\sigma = 0.01$ | $\sigma = 0.1$ | $7403 \pm 862$ | $8854 \pm 168$ | $6006 \pm 555$ | $7842 \pm 625$ | $4715 \pm 611$ | SID time-out |
| 9. | Dense root causes $\mathbf{C}$ | $p = 0.1$ | $p = 0.5$ | $6944 \pm 401$ | $8334 \pm 278$ | $5008 \pm 489$ | $7510 \pm 187$ | $4072 \pm 181$ | SID time-out |
| 10. | Fixed support | No | Yes | $6952 \pm 619$ | $7175 \pm 578$ | $7350 \pm 569$ | $8632 \pm 383$ | $\mathbf{5477 \pm 279}$ | SID time-out |

2005] dataset we perform an approximation of the underlying root causes. As the known matrix is unweighted we choose to assign them the weights that would minimize the sparsity of the root causes $\|\mathbf{C}\|_1$. The root causes $\mathbf{C}$ that are computed based on this optimization are depicted in Fig. 6a. In Fig. 6b We see the distribution of the values of root causes: there are many values with low magnitude close to zero and a few with significantly high magnitude. Also many at roughly the same location (Fig. 6a).

## B.6 Implementation details

**Training details.** In the implementation of SparceRC we initiate a PyTorch nn.Module with a single $d \times d$ weighted matrix represented as a linear layer. The model is trained so that the final result will have a weight matrix which approximates the original DAG adjacency matrix. To train our model we use the Adam optimizer [Kingma and Ba, 2014] with learning rate $10^{-3}$. We choose $\lambda = 10^{-3}$ as the coefficient for the $L^1$ regularizer of optimization problem (10). For NOTEARS we also choose $\lambda = 10^{-3}$. The rest of the methods are used with

Table 8: Runtime [s].

(a)

| | Hyperparameter | Default | Change | SparseRC (ours) | GOLEM | NOTEARS | DAGMA | DirectLiNGAM | PC | GES |
|---|---|---|---|---|---|---|---|---|---|---|
| 1. | Default settings | | | $10 \pm 1.8$ | $529 \pm 210$ | $796 \pm 185$ | $320 \pm 94$ | $106 \pm 1.5$ | $22 \pm 3.1$ | $7.1 \pm 1.5$ |
| 2. | Graph type | Erdös-Renyi | Scale-free | $11 \pm 1.1$ | $460 \pm 184$ | $180 \pm 7.2$ | $258 \pm 49$ | $108 \pm 0.8$ | $115 \pm 59$ | $5.2 \pm 1.0$ |
| 3. | $N_c, N_x$ distribution | Gaussian | Gumbel | $8.2 \pm 0.7$ | $349 \pm 125$ | $251 \pm 48$ | $256 \pm 70$ | $107 \pm 0.8$ | $15 \pm 2.1$ | $4.6 \pm 0.4$ |
| 4. | Edges / Vertices | 4 | 10 | $14 \pm 1.0$ | $347 \pm 121$ | $471 \pm 82$ | $217 \pm 11$ | $149 \pm 24$ | $5.6 \pm 1.1$ | $64 \pm 2.6$ |
| 5. | Samples | $n = 1000$ | $n = 100$ | $13 \pm 0.7$ | $194 \pm 9.6$ | $679 \pm 72$ | $254 \pm 28$ | error | $2.2 \pm 0.1$ | $3.9 \pm 0.5$ |
| 6. | Standardization | No | Yes | $11 \pm 1.9$ | $326 \pm 145$ | $781 \pm 76$ | $157 \pm 19$ | $107 \pm 0.6$ | $17 \pm 4.4$ | $5.3 \pm 1.4$ |
| 7. | Larger weights in $\mathbf{A}$ | $(0.1, 0.9)$ | $(0.5, 2)$ | $8.4 \pm 0.6$ | $431 \pm 177$ | $2834 \pm 228$ | $493 \pm 46$ | $117 \pm 3.2$ | $2.3 \pm 0.1$ | $21 \pm 4.7$ |
| 8. | $N_c, N_x$ deviation | $\sigma = 0.01$ | $\sigma = 0.1$ | $8.7 \pm 0.7$ | $309 \pm 63$ | $433 \pm 53$ | $257 \pm 37$ | $111 \pm 7.1$ | $19 \pm 4.8$ | $5.3 \pm 0.5$ |
| 9. | Dense root causes $\mathbf{C}$ | $p = 0.1$ | $p = 0.5$ | $9.1 \pm 0.7$ | $334 \pm 121$ | $427 \pm 35$ | $133 \pm 20$ | $142 \pm 3.6$ | $15 \pm 5.4$ | $4.9 \pm 1.2$ |
| 10. | Fixed support | No | Yes | $15 \pm 2.0$ | $360 \pm 142$ | $669 \pm 386$ | $501 \pm 91$ | $106 \pm 0.6$ | $5.3 \pm 3.2$ | $6.2 \pm 2.1$ |

(b)

| | Hyperparameter | Default | Change | LiNGAM | CAM | DAG-NoCurl | fGES | sortnregress | MMHC |
|---|---|---|---|---|---|---|---|---|---|
| 1. | Default settings | | | $\mathbf{2.4 \pm 0.0}$ | $372 \pm 7.4$ | $15 \pm 2.8$ | $7.9 \pm 2.7$ | $2.6 \pm 0.2$ | $3.5 \pm 0.3$ |
| 2. | Graph type | Erdös-Renyi | Scale-free | $2.7 \pm 0.0$ | $282 \pm 9.3$ | $13 \pm 1.1$ | $6.2 \pm 0.9$ | $\mathbf{2.0 \pm 0.0}$ | $210 \pm 160$ |
| 3. | $N_c, N_x$ distribution | Gaussian | Gumbel | $2.5 \pm 0.3$ | $312 \pm 38$ | $11 \pm 1.0$ | $6.2 \pm 1.4$ | $2.1 \pm 0.3$ | $3.4 \pm 0.6$ |
| 4. | Edges / Vertices | 4 | 10 | $3.1 \pm 0.5$ | $341 \pm 9.7$ | $24 \pm 3.8$ | $573 \pm 167$ | $\mathbf{2.1 \pm 0.0}$ | $3.1 \pm 0.2$ |
| 5. | Samples | $n = 1000$ | $n = 100$ | error | time-out | $33 \pm 4.9$ | error | error | $\mathbf{0.7 \pm 0.1}$ |
| 6. | Standardization | No | Yes | $2.4 \pm 0.1$ | $304 \pm 13$ | $13 \pm 1.5$ | $7.2 \pm 1.9$ | $\mathbf{2.0 \pm 0.0}$ | $3.7 \pm 0.6$ |
| 7. | Larger weights in $\mathbf{A}$ | $(0.1, 0.9)$ | $(0.5, 2)$ | $3.0 \pm 0.1$ | $339 \pm 10.0$ | $106 \pm 11$ | $118 \pm 46$ | $2.3 \pm 0.0$ | $2.5 \pm 0.2$ |
| 8. | $N_c, N_x$ deviation | $\sigma = 0.01$ | $\sigma = 0.1$ | $8.2 \pm 2.1$ | $323 \pm 2.8$ | $7.8 \pm 0.9$ | $4.4 \pm 0.8$ | $\mathbf{2.0 \pm 0.0}$ | $3.7 \pm 0.3$ |
| 9. | Dense root causes $\mathbf{C}$ | $p = 0.1$ | $p = 0.5$ | $6.8 \pm 0.7$ | $306 \pm 7.4$ | $11 \pm 3.1$ | $6.8 \pm 0.5$ | $\mathbf{2.0 \pm 0.0}$ | $3.1 \pm 0.2$ |
| 10. | Fixed support | No | Yes | $5.5 \pm 0.6$ | $342 \pm 11$ | $52 \pm 15$ | $6.1 \pm 5.1$ | $\mathbf{2.0 \pm 0.0}$ | $2.4 \pm 0.6$ |

Table 9: SparseRC weight reconstruction performance on larger DAGs.

| Nodes $d$, samples $n$ | Avg. $L^1$ loss | Max-$L^1$ loss | Avg. $L^2$ loss | NMSE |
|---|---|---|---|---|
| $d = 200$, $n = 500$ | 0.071 | 0.317 | 0.003 | 0.087 |
| $d = 500$, $n = 1000$ | 0.066 | 0.275 | 0.002 | 0.079 |
| $d = 1000$, $n = 5000$ | 0.050 | 0.287 | 0.001 | 0.060 |
| $d = 2000$, $n = 10000$ | 0.050 | 0.388 | 0.001 | 0.062 |
| $d = 3000$, $n = 10000$ | 0.054 | 0.399 | 0.001 | 0.067 |

default parameters, except for the post-processing threshold which is always set to $0.09$, except to the case of larger weights (Table 1, row 6) where it is set slightly below the weights $\omega = 0.4$.

**Resources.** Our experiments were run on a single laptop machine with 8 core CPU with 32GB RAM and an NVIDIA GeForce RTX 3080 GPU.

**Licenses.** We use code repositories that are open-source and publicly available on github. All repositories licensed under the Apache 2.0 or MIT license. In particular we use the github repositories of DAGMA [Bello et al., 2022] https://github.com/kevinsbello/dagma, GOLEM [Ng et al., 2020] https://github.com/ignavierng/golem, NOTEARS [Zheng et al., 2018] https://github.com/xunzheng/notears, DAG-NoCurl [Yu et al., 2021] https://github.com/fishmoon1234/DAG-NoCurl, LiNGAM [Shimizu et al., 2006] https://github.com/cdt15/lingam the implementation of sortnregress [Reisach et al., 2021] https://github.com/Scriddie/Varsortability and the causal discovery toolbox [Kalainathan and Goudet, 2019] https://github.com/FenTechSolutions/CausalDiscoveryToolbox. Our code is licensed under the MIT license and available publicly in github.

# C Global Minimizer

We provide the proof of Theorem 3.2 of the main paper. The data $\mathbf{X}$ are assumed to be generated via noise-less root causes as described . We denote by $\widehat{\mathbf{A}}$ the optimal solution of the optimization problem (8). We want to prove the following result.

**Theorem C.1.** *Consider a DAG with weighted adjacency matrix $\mathbf{A}$ with $d$ nodes. Given exponential (in $d$) number $n$ of samples $\mathbf{X}$ the matrix $\mathbf{A}$ is, with high probability, the global minimizer of the optimization problem (8).*

In particular what we will show the following result, from which our theorem follows directly.

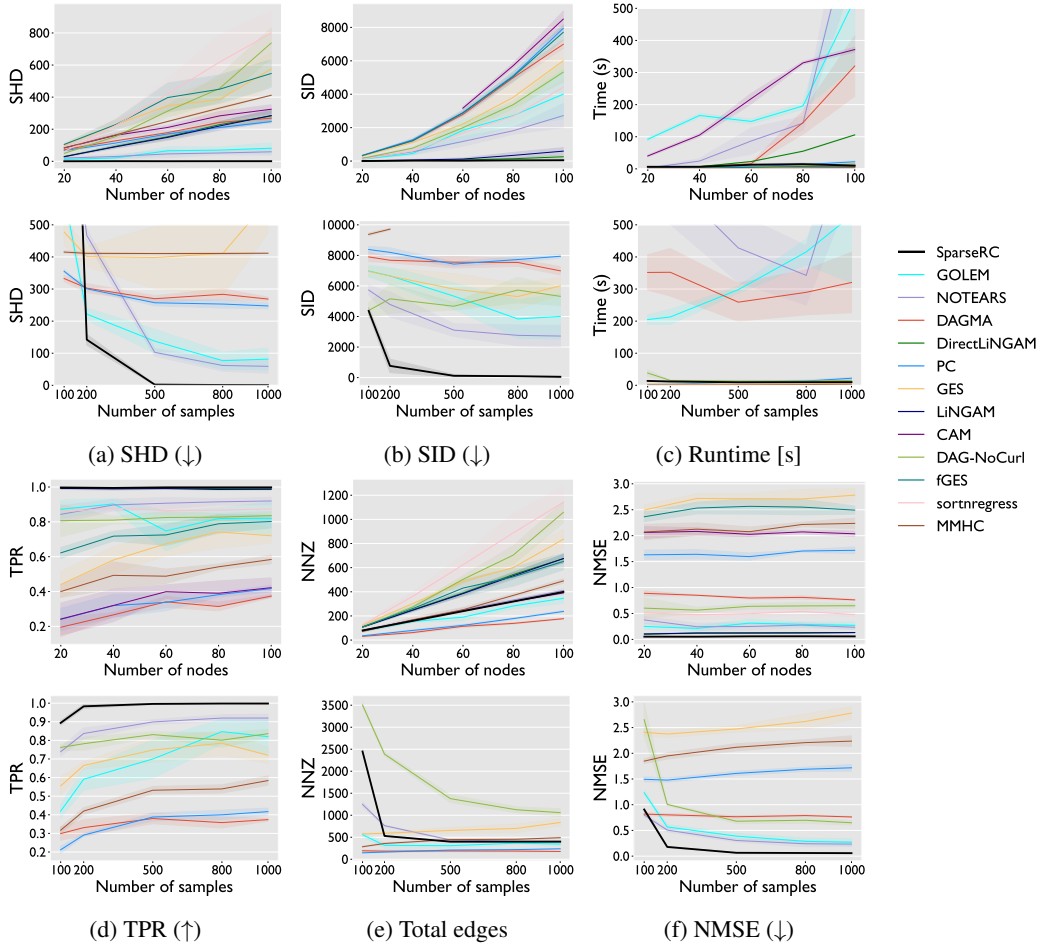

Figure 4: Plots illustrating performance metrics (a) SHD (lower is better), (b) SID (lower is better), (c) Time [seconds], (d) TPR (higher is better), (e) Total number of proposed edges and (f) NMSE (lower is better). Each metric is evaluated in two experimental scenarios: when varying the number of rows (upper figure) and when varying the number of samples (lower figure).

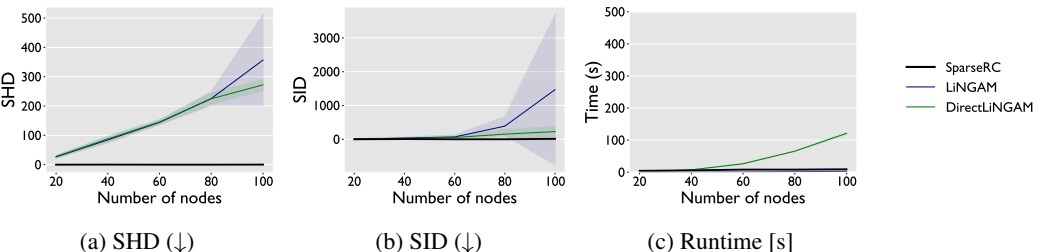

Figure 5: Evaluation of LiNGAM's performance when $\mathbf{N}_x = \mathbf{0}$.

**Theorem C.2.** *Consider a DAG with weighted adjacency matrix* $\mathbf{A}$. *Given that the number of data* $n$ *satisfies*

$$n \geq \frac{2^{3d-2}d(d-1)}{(1-\delta)^2 p^k (1-p)^{d-k}} \tag{18}$$

*where* $k = \lfloor dp \rceil$ *and*

$$\delta \geq \max \left\{ \frac{1}{p^k (1-p)^{d-k} \binom{d}{k}} \sqrt{\frac{1}{2} \ln \left( \frac{1}{\epsilon} \right)}, \binom{d}{k} \sqrt{\frac{1}{2} \ln \left( \frac{\binom{d}{k}}{\epsilon} \right)} \right\}. \tag{19}$$

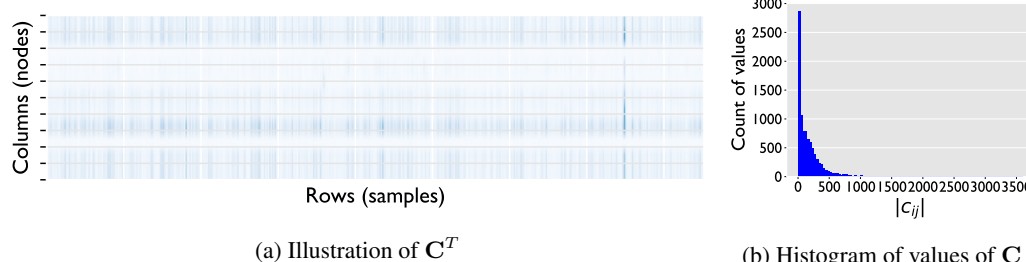

(a) Illustration of $\mathbf{C}^T$

(b) Histogram of values of $\mathbf{C}$

Figure 6: Root causes $\mathbf{C}$ in [Sachs et al., 2005] dataset. (a) Root causes illustration: darker color means higher value. (b) Histogram of values of $\mathbf{C}$.

*Then with probability $(1 - \epsilon)^2$ the solution to the optimization problem* (8) *coincides with the true DAG, namely* $\widehat{\mathbf{A}} = \mathbf{A}$.

*Remark C.3.* The reason we choose $k = \lfloor dp \rfloor$ is that because of the Bernoulli trials for the non-zero root causes, the expected value of the cardinality of the support will be exactly $\lfloor dp \rfloor$. Thus we expect to have more concentration on that value of $k$ which is something we desire, since in the proof we use all possible patterns with support equal to $k$.

*Proof sketch.* Given a large (exponential), but finite number of data $n$ we can find, with high likelihood (approaching one as $n$ increases), pairs of sub-matrices of root causes matrices $\mathbf{C}$ (true) and $\widehat{\mathbf{C}}$ (corresponding to the solution of optimization), with the same support and covering all possibilities of support with $k$ non-zero elements, due to the assumption of randomly varying support. Exploiting the support of the pairs we iteratively prove that the entries of the matrices $\mathbf{A}$ (true) and $\widehat{\mathbf{A}}$ (solution of the optimization) are equal. The complete proof is provided next. □

We begin with some important observations and definitions.

**Lemma C.4.** *If* $\overline{\widehat{\mathbf{A}}} = \overline{\mathbf{A}}$ *then* $\widehat{\mathbf{A}} = \mathbf{A}$.

*Proof.* We have that

$$\mathbf{I} + \overline{\widehat{\mathbf{A}}} = \mathbf{I} + \overline{\mathbf{A}} \Leftrightarrow \left(\mathbf{I} + \overline{\widehat{\mathbf{A}}}\right)^{-1} = \left(\mathbf{I} + \overline{\mathbf{A}}\right)^{-1} \Leftrightarrow \left(\mathbf{I} - \widehat{\mathbf{A}}\right) = \left(\mathbf{I} - \mathbf{A}\right) \Leftrightarrow \widehat{\mathbf{A}} = \mathbf{A} \tag{20}$$

□

**Definition C.5.** We define $\widehat{\mathbf{C}} = \mathbf{X} - \mathbf{X}\widehat{\mathbf{A}}$ the root causes corresponding to the optimal adjacency matrix.

**Definition C.6.** Let $S \subset \{1, 2, 3, ..., d\}$ be a set of indices. We say that a root causes vector $\mathbf{c} = (c_1, ..., c_d)$ has support $S$ if $c_i = 0$ for $i \in [d] \setminus S$.

**Definition C.7.** For a given support $S$ we consider the set $R \subset [n]$ of the rows of $\mathbf{C}$ that have support $S$. Then, $\mathbf{C}_R, \widehat{\mathbf{C}}_R$ denote the submatrices consisting of the rows with indices in $R$.

**Lemma C.8.** *For any row subset $R \subset [n]$ we have that* $\text{rank}\left(\widehat{\mathbf{C}}_R\right) = \text{rank}\left(\mathbf{C}_R\right)$

*Proof.* We have that

$$\widehat{\mathbf{C}}\left(\mathbf{I} + \overline{\widehat{\mathbf{A}}}\right) = \mathbf{X} = \mathbf{C}\left(\mathbf{I} + \overline{\mathbf{A}}\right) \tag{21}$$

Therefore, since both $\mathbf{A}, \widehat{\mathbf{A}}$ are acyclic and $\mathbf{I} + \overline{\widehat{\mathbf{A}}}, \mathbf{I} + \overline{\mathbf{A}}$ are invertible we have that

$$\widehat{\mathbf{C}}_R\left(\mathbf{I} + \overline{\widehat{\mathbf{A}}}\right) = \mathbf{C}_R\left(\mathbf{I} + \overline{\mathbf{A}}\right) \Rightarrow \text{rank}\left(\widehat{\mathbf{C}}_R\right) = \text{rank}\left(\mathbf{C}_R\right) \tag{22}$$

□

**Lemma C.9.** *For any row subset $R \subset [n]$ of root causes $\mathbf{C}$ with the same support $S$ such that $|S| = k$ and $|R| = r \geq k$ the nonzero columns of $\mathbf{C}_R$ are linearly independent with probability 1 and therefore $\text{rank}\left(\mathbf{C}_R\right) = k$.*

*Proof.* Assume $\mathbf{c}_1, ..., \mathbf{c}_k$ are the non-zero columns of $\mathbf{C}_R$. Then each $\mathbf{c}_i$ is a vector of dimension $r$ whose each entry is sampled uniformly at random in the range $(0, 1]$. Given any $k - 1$ vectors out of $\mathbf{c}_1, ..., \mathbf{c}_k$ their linear span can form a subspace of $(0, 1]^r$ of dimension at most $k - 1$. However, since every $\mathbf{c}_i$ is sampled uniformly at random from $(0, 1]^r$ and $r \geq k > k - 1$ the probability that $\mathbf{c}_i$ lies in the linear span of the other $k - 1$ vectors has measure $0$. Therefore, the required result holds with probability $1$. $\qquad\square$

**Lemma C.10.** *With probability $(1 - \varepsilon)^2$ for every different support $S$ with $|S| = k$ there are rows $R$ such that $\mathbf{C}_R$ have support $S$ and*

$$|R| > 2^{3d-2}d(d - 1) \tag{23}$$

*Proof.* Set $l = 2^{3d-2}d(d - 1)$ and $K = \frac{l\binom{d}{k}}{(1-\delta)}$. Also set $N$ be the random variable representing the number of root causes $\mathbf{c}$ with $|supp(\mathbf{c})| = k$ and $N_i$ the number of repetitions of the $i-$th $k-$support pattern, $i = 1, ..., \binom{d}{k}$.

We first use conditional probability.

$$\mathbb{P}\left(N \geq K \cap N_i \geq l, \forall i\right) = \mathbb{P}\left(N_i \geq l, \forall i | N \geq k\right)\mathbb{P}\left(N \geq K\right) \tag{24}$$

Now we will show that $\mathbb{P}\left(N \geq K\right) \geq (1 - \epsilon)$ and $\mathbb{P}\left(N_i \geq l, \forall i | N \geq k\right) \geq (1 - \epsilon)$ using Chernoff bounds. From Hoeffding inequality we get:

$$\mathbb{P}\left(N \leq (1 - \delta)\mu\right) \leq e^{-2\delta^2\mu^2/n^2} \tag{25}$$

where $\mu = np^k(1 - p)^{d-k}\binom{d}{k}$ is the expected value of $N$, $\delta \geq \frac{1}{p^k(1-p)^{d-k}\binom{d}{k}}\sqrt{\frac{1}{2}\ln\left(\frac{1}{\epsilon}\right)}$ and $n \geq \frac{1}{1-\delta}\frac{K}{p^k(1-p)^{d-k}\binom{d}{k}}$ so

$$\mathbb{P}\left(N \leq K\right) \leq \mathbb{P}\left(N \leq (1 - \delta)\mu\right) \leq e^{-2\delta^2\mu^2/n^2} \leq \varepsilon \Leftrightarrow \mathbb{P}\left(N \geq K\right) \geq 1 - \epsilon \tag{26}$$

Now given that we have at least $K$ root causes with support exactly $k$, we will show that, on exactly $K$ such root causes, each different pattern will appear at least $l$ times with probability $(1 - \epsilon)$. From the union bound

$$\mathbb{P}\left(\bigcup_i N_i \leq l\right) \leq \sum_i \mathbb{P}\left(N_i \leq l\right) \tag{27}$$

So we need to show that $\mathbb{P}\left(N_i \leq l\right) \leq \frac{\epsilon}{\binom{d}{k}}$. We use again the Hoeffding inequality.

$$\mathbb{P}\left(N_i \leq (1 - \delta)\mu\right) \leq e^{-2\delta^2\mu^2/K^2} \tag{28}$$

where the expected value is $\mu = \frac{K}{\binom{d}{k}} = \frac{l}{1-\delta}$ since now the probability is uniform over all possible $k-$sparsity patterns. Given $\delta \geq \binom{d}{k}\sqrt{\frac{1}{2}\ln\left(\frac{\binom{d}{k}}{\epsilon}\right)}$ we derive

$$\mathbb{P}\left(N_i \leq l\right) = \mathbb{P}\left(N_i \leq (1 - \delta)\mu\right) \leq e^{-2\delta^2\mu^2/K^2} \leq \frac{\epsilon}{\binom{d}{k}} \tag{29}$$

The result follows. $\qquad\square$

**Lemma C.11.** *Assume that the data number $n$ is large enough so that for every different support $S$ we have the rows $R$ such that $\mathbf{C}_R$ have support $S$ and $|R| > 2^d l$ where*

$$l > 2^{2d-2}d(d - 1) = 2^d \sum_{k=2}^d \binom{d}{k}k(k - 1) \tag{30}$$

*Then there exist rows $\hat{R}$ such that $|\hat{R}| > k$, $\mathbf{C}_{\hat{R}}$ has support $S$ and $\widehat{\mathbf{C}}_{\hat{R}}$ has support $\hat{S}$ with $|\hat{S}| = k$. Moreover, both the non-zero columns of $\mathbf{C}_{\hat{R}}, \widehat{\mathbf{C}}_{\hat{R}}$ form two linearly independent set of vectors. In words, this means that the data are many enough so we can always find pairs of corresponding submatrices of root causes with each consisting of $k$ non-zero columns.*

*Proof.* According to the assumption there exists a row-submatrix $\mathbf{C}_R$ where all rows have support $R$ and $|R| > 2^d l$. Group the root causes $\widehat{\mathbf{C}}_R$ into all different supports, which are $2^d$ in number. Take any such group $R' \subset R$ of rows of $\widehat{\mathbf{C}}$ that all have the same support $S'$. If $|R'| \geq k$ then from Lemma C.8 rank $\left(\widehat{\mathbf{C}}_{R'}\right) =$ rank $\left(\mathbf{C}_{R'}\right) = k$ so $\widehat{\mathbf{C}}_{R'}$ will have at least $k$ non-zero columns, and since the support is fixed, it will have at least $k$ non-zero elements at each row, which means at least as many non-zeros as $\mathbf{C}_{R'}$. Therefore $\widehat{\mathbf{C}}_{R'}$ can only

have less non-zero elements if $|R'| < k$, and in that case $\mathbf{C}_{R'}$ has at most $k(k-1)$ more elements. If we count for all $k = 1, ..., d$ all different supports of $\widehat{\mathbf{C}}_{R'}$ for all possible supports of $\mathbf{C}_R$ this gives that $\widehat{\mathbf{C}}$ can have at most $\sum_{k=2}^{d} \binom{d}{k} 2^d k(k-1)$ less non-zero elements compared to $\mathbf{C}$.

Due to the pigeonhole principle, there exists $\hat{R} \subset R$ and $|\hat{R}| > l$ with $\widehat{\mathbf{C}}_{\hat{R}}$ all having the same support $\hat{S}$, not necessarily equal to $S$. According to our previous explanation we need to have at least $k$ non-zero columns in $\widehat{\mathbf{C}}_{\hat{R}}$. If we had $k+1$ columns then this would give $l$ more non-zero elements, but

$$l > 2^{2d-2}d(d-1) = 2^d d(d-1)2^{d-2} = 2^d \sum_{k=0}^{d-2} \binom{d-2}{k-2} d(d-1) = 2^d \sum_{k=2}^{d} \binom{d}{k} k(k-1) \quad (31)$$

So then $\|\widehat{\mathbf{C}}\|_0 > \|\mathbf{C}\|$ which is a contradiction due to the optimality of $\widehat{\mathbf{A}}$. Therefore, $\widehat{\mathbf{C}}_{\hat{R}}$ has exactly $k$ non-zero columns which necessarily need to be linearly independent in order to have $\text{rank}\left(\widehat{\mathbf{C}}_{\hat{R}}\right) = \text{rank}\left(\mathbf{C}_{\hat{R}}\right) = k$ as necessary. The linear independence of the columns of $\mathbf{C}_{\hat{R}}$ follows from Lemma C.9 since $l \gg k$. $\qquad\square$

**Definition C.12.** A pair $\mathbf{C}_R, \widehat{\mathbf{C}}_R$ constructed according to Lemma C.11 that have fixed support $S, \hat{S}$ respectively with cardinality $k$ each and $|R|$ large enough so that $\text{rank}\left(\widehat{\mathbf{C}}_R\right) = \text{rank}\left(\mathbf{C}_R\right) = k$, will be called a $k-pair$ of submatrices.

*Remark* C.13. For notational simplicity we will drop the index $R$ whenever the choice of the rows according to the sparsity pattern $S$ is defined in the context.

The complete Theorem 3.2 follows after combining the following two propositions C.16 and C.18. Before, the proof of the first proposition we need the following lemmas.

**Lemma C.14.** $\mathbf{A}$ *is (strictly) upper triangular if and only if* $\overline{\mathbf{A}}$ *is (strictly) upper triangular.*

*Proof.* If $\mathbf{A}$ is (strictly) upper triangular then it is straightforward to show the same for $\overline{\mathbf{A}} = \mathbf{A} + \mathbf{A}^2 + ... + \mathbf{A}^{d-1}$. For the other way round, remember that the inverse of an upper triangular matrix (if exists) it is an upper triangular matrix. However, the inverse of $\mathbf{I} + \overline{\mathbf{A}}$ is $\mathbf{I} - \mathbf{A}$ which means, that if $\overline{\mathbf{A}}$ is strictly upper triangular, then $\mathbf{I} + \overline{\mathbf{A}}$ and so is $\mathbf{I} - \mathbf{A}$ are upper triangular which in turn reduces to $\mathbf{A}$ being strictly upper triangular. $\qquad\square$

**Lemma C.15.** *Consider* $\mathbf{C}, \widehat{\mathbf{C}}$ *a* $k-pair$. *Also let* $\mathbf{X}$ *be the corresponding submatrix of data. If* $\mathbf{X}_{:,i} = \mathbf{0}$ *then*

$$\widehat{\mathbf{C}}_{:,i} = \mathbf{0} \text{ and } \overline{\hat{a}}_{ji} = 0 \quad \forall j \in supp\left(\widehat{\mathbf{C}}\right) \quad (32)$$

*Proof.*

$$\mathbf{0} = \mathbf{X}_{:,i} = \sum_{j=1}^{d} \overline{\hat{a}}_{ji} \widehat{\mathbf{C}}_{:,j} + \widehat{\mathbf{C}}_{:,i} = \sum_{j \in \text{supp}(\widehat{\mathbf{C}})} \overline{\hat{a}}_{ji} \widehat{\mathbf{C}}_{:,j} + \widehat{\mathbf{C}}_{:,i} \quad (33)$$

If $\widehat{\mathbf{C}}_{:,i} \neq \mathbf{0}$ then $i \in \text{supp}\left(\widehat{\mathbf{C}}\right)$, and the expression above constitutes a linear combination of the support columns with not all coefficients non-zero. This contradicts Lemma C.9. Thus $\mathbf{C}_{:,i} = \mathbf{0}$ and $\overline{\hat{a}}_{ji} = 0 \quad \forall j \in \text{supp}\left(\widehat{\mathbf{C}}\right)$. $\qquad\square$

We are now ready to prove Prop. C.16.

**Proposition C.16.** *If the data* $\mathbf{X}$ *are indexed such that* $\mathbf{A}$ *is (strictly) upper triangular, then so is* $\widehat{\mathbf{A}}$.

*Remark* C.17. Based on lemma C.14, in our proof of Prop. C.16 next we may only care about the transitive weights, namely we will only need to show that $\overline{\hat{a}}_{ji} = 0$ for all $i < j$.

*Proof.* We first choose a $k-pair$ $\mathbf{C}, \widehat{\mathbf{C}}$ such that the support $S$ of $\mathbf{C}$ is concetrated to the last $k$ columns, namely $\mathbf{C}_{:,i} = 0$ for $i = 1, ..., d - k$. Then the corresponding data submatrix $\mathbf{X}$ will necessarily have the same sparsity pattern since the values are computed according to predecessors, which in our case lie in smaller indices. Thus $\mathbf{X}_{:,i} = \mathbf{0} \quad \forall i = 1, ..., d - k$ and necessarily $\widehat{\mathbf{C}}$ will follow the same sparsity pattern. Moreover, according to Lemma C.15 we get that

$$\overline{\hat{a}}_{ji} = 0, \text{ for all } 1 \leq i \leq d - k \text{ and } d - k + 1 \leq j \leq d. \quad (34)$$

We notice that the desired condition is fullfilled for $i = d - k$, i.e. it has zero influence from a node with larger index. We now prove the same sequentially for $i = d - k - 1, d - k - 2, ..., 1$ by moving each time (one step) left the leftmost index of the support $S$. We prove the following statement by induction:

$$P(l) : \overline{\hat{a}}_{ji} = 0 \text{ for } l < j, i < j, i \leq d - k \quad (35)$$

We know that $P(d-k)$ is true and $P(1)$ gives the required relation for all indices $i = 1, ..., d-k$. Now we assume that $P(l)$ holds. If we pick $k-$pair $\mathbf{C}, \widehat{\mathbf{C}}$ such that $\mathbf{C}$ has support $S = \{l, d-k+2, d-k+3..., d\}$. Then $\mathbf{X}_{:,i} = \mathbf{0}$ for all $i < l$ which with Lemma C.15 gives $\widehat{\mathbf{C}}_{:,i} = \mathbf{0}$ for $i < l$ which means $\text{supp}\left(\widehat{\mathbf{C}}\right) \subset \{l, l+1, ..., d\}$ and $\overline{\overline{a}}_{ji} = 0$ for $j \in \text{supp}\left(\widehat{\mathbf{C}}\right)$. Note, that by the induction hypothesis we have that $\overline{\overline{a}}_{jl} = 0$ for all $l < j$. However it is true that $\mathbf{X}_{:,l} = \mathbf{C}_{:,l}$ and also

$$\mathbf{X}_{:,l} = \sum_{j \in \text{supp}(\widehat{\mathbf{C}})}^{d} \overline{\overline{a}}_{jl} \widehat{\mathbf{C}}_{:,j} + \widehat{\mathbf{C}}_{:,l} = \widehat{\mathbf{C}}_{:,l} \tag{36}$$

Therefore $l \in \text{supp}\left(\widehat{\mathbf{C}}\right)$ and thus $\overline{\overline{a}}_{li} = 0$ for all $i < l$ which combined with $P(l)$ gives

$$\overline{\overline{a}}_{ji} = 0 \text{ for } l-1 < j, \ i < j, \ i \le d-k \tag{37}$$

which is exactly $P(l-1)$ and the induction is complete.

Now it remains to show that $\overline{\overline{a}}_{ji} = 0$ for $d-k+1 \le i \le d$ and $i < j$. We will again proceed constructively using induction. This time we will sequentially choose support $S$ that is concentrated on the last $k+1$ columns and at each step we will move the zero column one index to the right. For $l = d-k+1, ..., d$ let's define:

$$Q(l) : \ \overline{\overline{a}}_{jl} = 0 \text{ for } l < j \text{ and } \overline{\overline{a}}_{jl} = \overline{a}_{jl} \text{ for } d-k \le j < l \tag{38}$$

First we show the base case $Q(d-k+1)$. For this we choose a $k-$pair $\mathbf{C}, \widehat{\mathbf{C}}$ such that $\mathbf{C}$ has support $S = \{d-k, d-k+2, d-k+3, ..., d\}$. It is true that $\mathbf{X}_{:,i} = \mathbf{0}$ for $i = 1, ..., d-k-1$ hence $\widehat{\mathbf{C}}_{:,i} = \mathbf{0}$ for $i \le d-k-1$ and therefore the node $d-k$ doesn't have any non-zero parents, since also $\overline{\overline{a}}_{j(d-k)} = 0$ for $d-k < j$ from previous claim. Therefore $\widehat{\mathbf{C}}_{:,d-k} = \mathbf{X}_{:,d-k} = \mathbf{C}_{:,d-k}$. Also, for $l = d-k+1$

$$\mathbf{X}_{:,l} = \overline{a}_{d-k,l} \mathbf{C}_{:,d-k} = \overline{a}_{d-k,l} \widehat{\mathbf{C}}_{:,d-k} \tag{39}$$

The equation from $\widehat{\mathbf{A}}$ gives:

$$\mathbf{X}_{:,l} = \sum_{j=d-k}^{d} \overline{\overline{a}}_{jl} \widehat{\mathbf{C}}_{:,j} + \widehat{\mathbf{C}}_{:,l} \Rightarrow \left(\overline{a}_{d-k,l} - \overline{\overline{a}}_{d-k,l}\right) \widehat{\mathbf{C}}_{:,d-k} + \sum_{j=d-k+1}^{d} \overline{\overline{a}}_{jl} \widehat{\mathbf{C}}_{:,j} + \widehat{\mathbf{C}}_{:,l} = \mathbf{0} \tag{40}$$

From the linear Independence Lemma C.9 of the support of $\widehat{\mathbf{C}}$ we necessarily need to have $\widehat{\mathbf{C}}_{:,l} = \mathbf{0}, \overline{a}_{d-k,l} = \overline{\overline{a}}_{d-k,l}$ and $\overline{\overline{a}}_{jl} = 0$ for $l < j$ which gives the base case.

For the rest of the induction we proceed in a similar manner. We assume with strong induction that all $Q(d-k+1), ..., Q(l)$ are true and proceed to prove $Q(l+1)$. Given these assumptions we have that

$$\overline{\overline{a}}_{ji} = \overline{a}_{ji} \text{ for } d-k \le j < i, \ d-k \le i \le l \text{ and } \overline{\overline{a}}_{ji} = 0 \text{ for } i < j, \ d-k \le i \le l \tag{41}$$

Consider root causes support $S = \{d-k, d-k+1, ..., l, l+2, ..., d\}$ (the $(l+1)-$th column is $\mathbf{0}$) for the $k-$pair $\mathbf{C}, \widehat{\mathbf{C}}$. Then we have the equations:

$$\mathbf{X}_{:,d-k} = \mathbf{C}_{:,d-k} = \widehat{\mathbf{C}}_{:,d-k}$$

$$\mathbf{X}_{:,d-k+1} = \overline{a}_{d-k,d-k+1} \mathbf{C}_{:,d-k} + \mathbf{C}_{:,d-k+1} = \overline{\overline{a}}_{d-k,d-k+1} \widehat{\mathbf{C}}_{:,d-k} + \widehat{\mathbf{C}}_{:,d-k+1} \Rightarrow \widehat{\mathbf{C}}_{:,d-k+1} = \mathbf{C}_{:,d-k+1}$$

$$\vdots$$

$$\mathbf{X}_{:,l} = \sum_{j=d-k}^{l-1} \overline{a}_{jl} \mathbf{C}_{:,j} + \mathbf{C}_{:,l} = \sum_{j=d-k}^{l-1} \overline{\overline{a}}_{jl} \widehat{\mathbf{C}}_{:,j} + \widehat{\mathbf{C}}_{:,l} \Rightarrow \widehat{\mathbf{C}}_{:,l} = \mathbf{C}_{:,l}$$

Where we used the linear independence lemma and sequentially proved that the root causes columns up to $l$ are equal. The equation for the $(l+1)-$th column now becomes:

$$\mathbf{X}_{:,l+1} = \sum_{j=d-k}^{l} \overline{a}_{j,l+1} \mathbf{C}_{:,j} + \mathbf{C}_{:,l} = \sum_{j=d-k}^{d} \overline{\overline{a}}_{j,l+1} \widehat{\mathbf{C}}_{:,j} + \widehat{\mathbf{C}}_{:,l+1} \tag{42}$$

$$\Leftrightarrow \sum_{j=d-k}^{d} \left(\overline{\overline{a}}_{j,l+1} - \overline{a}_{j,l+1}\right) \widehat{\mathbf{C}}_{:,j} + \widehat{\mathbf{C}}_{:,l+1} = \mathbf{0} \Rightarrow \begin{cases} \overline{\overline{a}}_{j,l+1} = 0 \text{ for } l+1 < j \\ \overline{\overline{a}}_{j,l+1} = \overline{a}_{j,l+1} \text{ for } j < l+1 \\ \widehat{\mathbf{C}}_{:,l+1} = \mathbf{0} \end{cases} \tag{43}$$

where the last set of equalities follows from linear independence. This concludes the induction and the proof. $\qquad \square$

To complete the proof of Theorem 3.2 it remains to show the following proposition.

**Proposition C.18.** *If both* $\mathbf{A}, \widehat{\mathbf{A}}$ *are upper triangular then* $\widehat{\mathbf{A}} = \mathbf{A}$.

For our proof we use the following definition.

**Definition C.19.** We denote by $\mathcal{P}_k$ the set of all $k-$pairs $\mathbf{C}_R, \widehat{\mathbf{C}}_R$ for all possible support patterns.

Now we proceed to the proof.

*Proof.* We will show equivalently that $\overline{\widehat{\mathbf{A}}} = \overline{\mathbf{A}}$ using two inductions. First we show for $l = 1, ..., k$ the following statement.

$P(l)$ : For all $k-$pairs $\mathbf{C}, \widehat{\mathbf{C}}$ in $\mathcal{P}_k$ the first $l$ non-zero columns $\mathbf{C}_{:,i_1}, \mathbf{C}_{:,i_2}, ..., \mathbf{C}_{:,i_l}$ and $\widehat{\mathbf{C}}_{:,\hat{i}_1}, \widehat{\mathbf{C}}_{:,\hat{i}_2}, ..., \widehat{\mathbf{C}}_{:,\hat{i}_l}$ are in the same positions, i.e. $i_j = \hat{i}_j$ and

- either they are respectively equal $\mathbf{C}_{:,i_j} = \widehat{\mathbf{C}}_{:,i_j}$
- or $\mathbf{C}_{:,i_l}$ is in the last possible index, namely $i_l = d - (l-1)$

For the base case $P(1)$, consider a $k-$pair $\mathbf{C}, \widehat{\mathbf{C}}$ and let $i_1$ be the position of the first non-zero root causes column of $\mathbf{C}$. Then $\mathbf{X}_{:,i} = \mathbf{0}$ for $i < i_1$ and therefore from Lemma C.15 $\widehat{\mathbf{C}}_{:,i} = \mathbf{0}$ for $i < i_1$. Hence

$$\mathbf{X}_{:,i_1} = \sum_{j < i_1} \overline{\hat{a}}_{j i_1} \widehat{\mathbf{C}}_{:,j} + \widehat{\mathbf{C}}_{:,i_1} = \widehat{\mathbf{C}}_{:,i_1} \tag{44}$$

Therefore $\widehat{\mathbf{C}}_{:,i_1} = \mathbf{C}_{:,i_1}$ and we proved $P(1)$, by satisfying both the positioning and the first requirement.

Assuming now that $P(l)$ holds, we will show $P(l+1)$. Take any $k-$pair of $\mathcal{P}_k$ which we denote by $\mathbf{C}, \widehat{\mathbf{C}}$. Then, if $\mathbf{C}_{:,i_l}$ is in the last possible position, then necessarily $\mathbf{C}_{:,i_{l+1}}$ is in the last possible position. Moreover, from the induction hypothesis the first $l$ root causes columns are in the same positions. Therefore in the same manner, $\widehat{\mathbf{C}}_{:,i_l}$ is in the last position and $\widehat{\mathbf{C}}_{:,i_{l+1}}$, too. This case fulfills the desired statement.

If $\mathbf{C}_{:,i_l}$ is not in the last position, then from induction hypothesis, the first $l$ root causes columns are equal. If $\mathbf{C}_{:,i_{l+1}}$ is in the last position and the same holds $\widehat{\mathbf{C}}_{:,\hat{i}_{l+1}}$, the requirement is satisfied. Otherwise $\hat{i}_{l+1} < i_{l+1}$ and the equation for $\hat{i}_{l+1}$ gives:

$$\mathbf{X}_{:,\hat{i}_{l+1}} = \sum_{j=1}^{l} \overline{\hat{a}}_{i_j,\hat{i}_{l+1}} \widehat{\mathbf{C}}_{:,i_j} + \widehat{\mathbf{C}}_{:,\hat{i}_{l+1}} = \sum_{j=1}^{l} \overline{a}_{i_j,\hat{i}_{l+1}} \mathbf{C}_{:,i_j} + \mathbf{0} \Leftrightarrow \sum_{j=1}^{l} \left( \overline{\hat{a}}_{i_j,\hat{i}_{l+1}} - \overline{a}_{i_j,\hat{i}_{l+1}} \right) \widehat{\mathbf{C}}_{:,i_j} + \widehat{\mathbf{C}}_{:,\hat{i}_{l+1}} = \mathbf{0}$$

According to linear independence Lemma C.9 we necessarily derive $\widehat{\mathbf{C}}_{:,\hat{i}_{l+1}} = \mathbf{0}$, absurd. Thus $\hat{i}_{l+1} = i_{l+1}$ and the induction statement is fullfilled in that case.

It remains to consider the case where $\mathbf{C}_{:,i_{l+1}}$ is not in the last position. Since, $i_{l+1}$ is not the last position there exists a $k-$pair $\mathbf{C}', \widehat{\mathbf{C}}'$ such that the column $i_{l+1}$ is zero and the $(l+1)-$ root causes column of $\mathbf{C}'$ lies at $i'_{l+1} > i_{l+1}$. The equation at $i_{l+1}$ for $\mathbf{X}'$ gives:

$$\mathbf{X}'_{:,i_{l+1}} = \sum_{j=1}^{l} \overline{a}_{i_j,i_{l+1}} \mathbf{C}'_{:,i_j} = \sum_{j=1}^{l} \overline{\hat{a}}_{i_j,\hat{i}_{l+1}} \widehat{\mathbf{C}}'_{:,i_j} + \widehat{\mathbf{C}}'_{:,\hat{i}_{l+1}} \tag{45}$$

For the pair $\mathbf{C}', \widehat{\mathbf{C}}'$ the induction hypothesis holds, thus we derive:

$$\sum_{j=1}^{l} \left( \overline{\hat{a}}_{i_j,\hat{i}_{l+1}} - \overline{a}_{i_j,i_{l+1}} \right) \widehat{\mathbf{C}}'_{:,i_j} + \widehat{\mathbf{C}}'_{:,\hat{i}_{l+1}} = \mathbf{0} \tag{46}$$

Therefore, Lemma C.9 gives $\overline{\hat{a}}_{i_j,\hat{i}_{l+1}} = \overline{a}_{i_j,i_{l+1}}$ for all $j = 1, ..., l$ and returning back to the equation for $\mathbf{X}_{:,i_{l+1}}$ we derive:

$$\mathbf{X}_{:,i_{l+1}} = \sum_{j=1}^{l} \overline{a}_{i_j,i_{l+1}} \mathbf{C}_{:,i_j} + \mathbf{C}_{:,i_{l+1}} = \sum_{j=1}^{l} \overline{\hat{a}}_{i_j,\hat{i}_{l+1}} \widehat{\mathbf{C}}_{:,i_j} + \widehat{\mathbf{C}}_{:,\hat{i}_{l+1}} \tag{47}$$

$$\overset{P(l)}{=} \sum_{j=1}^{l} \overline{a}_{i_j,\hat{i}_{l+1}} \mathbf{C}_{:,i_j} + \widehat{\mathbf{C}}_{:,\hat{i}_{l+1}} \tag{48}$$

$$\Rightarrow \widehat{\mathbf{C}}_{:,\hat{i}_{l+1}} = \mathbf{C}_{:,\hat{i}_{l+1}} \tag{49}$$

which completes the induction step. Notice that $P(k)$ gives that for all $k-$pairs, the $k$ root causes columns are in the same position. Given this fact we will now show that $\widehat{\mathbf{A}} = \mathbf{A}$. We will prove by induction that for $l = k - 1, ..., 1, 0$

$$Q(l) : \overline{\widehat{a}}_{ij} = \overline{a}_{ij} \text{ for } 1 \leq i < j < d - l \tag{50}$$

To prove $Q(k-1)$ we choose all the $k-$pairs $\mathbf{C}, \widehat{\mathbf{C}}$, such that $\mathbf{C}_{:,i_2}$ is in the last possible position, $i_2 = d-k+2$. Then for $i_1 \leq d - k$ the columns $\mathbf{C}_{:,i_1}, \widehat{\mathbf{C}}_{:,i_1}$ lie at the same position and are equal. Choosing $i_1 = i$ and computing the equation for $\mathbf{X}_{:,j}$ where $i < j \leq d - k + 1$ gives:

$$\mathbf{X}_{:,j} = \overline{a}_{ij}\mathbf{C}_{:,i} = \overline{\widehat{a}}_{ij}\widehat{\mathbf{C}}_{:,i} = \overline{\widehat{a}}_{ij}\mathbf{C}_{:,i} \tag{51}$$

Therefore $\overline{\widehat{a}}_{ij} = \overline{a}_{ij}$ for all $1 \leq i < j \leq d - (k - 1)$ and $Q(k - 1)$ is satisfied. Next, assume that $Q(k - l)$ is true. We want to show $Q(k - l - 1)$. Similarly to the base case we consider all $k-$pairs such that $\mathbf{C}_{:,l+2}$ lies in its last possible position $i_{l+2} = d - k + l + 2$, and $i_{l+1} \leq d - k + l$. Since the $(l + 1)-$th column is not in the last position, from the previous induction we have that:

$$\begin{cases} \widehat{\mathbf{C}}_{:,i_1} = \mathbf{C}_{:,i_1} \\ \widehat{\mathbf{C}}_{:,i_2} = \mathbf{C}_{:,i_2} \\ \vdots \\ \widehat{\mathbf{C}}_{:,i_{l+1}} = \mathbf{C}_{:,i_{l+1}} \end{cases} \tag{52}$$

The equation for $d - k + l + 1$ gives:

$$\mathbf{X}_{:,d-k+l+1} = \sum_{j=1}^{l+1} \overline{a}_{i_j,d-k+l+1}\mathbf{C}_{:,i_j} \tag{53}$$

$$= \sum_{j=1}^{l+1} \overline{\widehat{a}}_{i_j,d-k+l+1}\widehat{\mathbf{C}}_{:,i_j} \tag{54}$$

$$= \sum_{j=1}^{l+1} \overline{\widehat{a}}_{i_j,d-k+l+1}\mathbf{C}_{:,i_j} \tag{55}$$

$$\xRightarrow{\text{Lemma C.9}} \overline{\widehat{a}}_{i_j,d-k+l+1} = \overline{a}_{i_j,d-k+l+1} \tag{56}$$

By choosing all such possible $k-$pairs the indices $i_j$ span all possibilities $1 \leq i < d - k + l + 1$. Combining this with $Q(k - l)$ we get that $Q(k - l - 1)$ is true and the desired result follows. $\qquad \square$

