# OpenReview forum: "Learning DAGs from Data with Few Root Causes"
_NeurIPS.cc/2023/Conference — NeurIPS 2023 poster_

### Official Review · Reviewer_Aevs · 2023-06-20

**Soundness:** 3 good
**Presentation:** 3 good
**Contribution:** 3 good
**Rating:** 6
**Confidence:** 4

**Summary:**

The paper studies a new causal discovery method, in which it assumes that the DAG data is produced by few data-generating events whose effect percolates through the DAG. They propose a simple but effective method to learn the true DAG based on the few roots assumption. The proposed method outperforms baselines in various settings.

**Strengths:**

1. The few roots assumption is reasonable. And the paper motivates it well.
2. The proposed solution is simple and effective.
3. The paper conducts experiments on both synthetic and real-world datasets, indicating the effectiveness of the proposed method.


**Weaknesses:**

1. It is not clear why the objective function Eq.(10) contains noise. It needs more clear explanation and derivation here.
2. I wonder whether the proposed method could find the root nodes at the same time rather than just learning the DAG.
3. The proposed method could not achieve the best results on real-world datasets. Hence, I doubt the few roots assumptions satisfy the real-world scenarios. BTW, it is better to say the network is a protein network rather than a gene network.


**Questions:**

Please reply to the questions in the weaknesses. I am willing to raise the score if all the concerns are solved.

**Limitations:**

Yes

---

> ### Author Rebuttal · Authors · 2023-08-09
>
> # Response to reviewer Aevs
>
> ## Weaknesses
>
> **Optimization robustness to noise.** The optimization problem doesn't contain noise explicitly. It is the convex  $L^1$ relaxation of the noise-free version of the optimization problem. Doing this relaxation allows some robustness to (low magnitude) noise as we later confirm in our experiments (Table 1, row 7).
>
> The objective in Eq. (10) only contains noise implicitly, after rewriting it with Eq. (4).
> If we have found the true adjacency matrix $\mathbf{A}$, the objective becomes
> $$\left\|\left\|\mathbf{X}\left(\mathbf{I} + \overline{\mathbf{A}}\right)^{-1}\right\|\right\| =\left\|\left\|\mathbf{C} + \mathbf{N}_c + \mathbf{N}_x\left(\mathbf{I} + \overline{\mathbf{A}}\right)^{-1}\right\|\right\| = \left\|\left\|\mathbf{C} + \mathbf{N}_c + \mathbf{N}_x\left(\mathbf{I} -\mathbf{A}\right)\right\|\right\|$$
> This means that we approximate the root causes up to noise (also in next answer).
>
> We will explain better.
>
> **Learning the root nodes.** Yes, we can do this in the following way. If we recover the true adjacency matrix $\mathbf{A}$ via optimization Eq. (10),
> we may compute an approximation $\widehat{\mathbf{C}}$ of the root causes $\mathbf{C}$ up to noise, by solving Eq. (4): $$\widehat{\mathbf{C}}= \mathbf{C} + \mathbf{N}_c + \mathbf{N}_x\left(\mathbf{I} -\mathbf{A}\right) = \mathbf{X}\left(\mathbf{I} + \overline{\mathbf{A}}\right)^{-1}.$$
> This obviously requires a very good (weighted) estimate of the original adjacency matrix. We evaluate the top-performing methods on the recovery of the root causes (and the associated values) with an additional experiment in Fig. 3 of the attached pdf.
>
> **Real-world dataset performance.** In Fig. 4 of the attached pdf we include an experiment that illustrates the sparsity in the root causes for the particular dataset [Sachs et. al., 2005]. Please see our general reply for further explanation and also our success in a causal discovery competition with real-world data.

---

> > ### Comment · Reviewer_Aevs · 2023-08-14
> > **Thanks for the rebuttal**
> >
> > Thanks to the authors for the rebuttal. The additional results have solved my concerns. I hope the authors could include the new results in the final version. I would like to raise my score to reflect the changes.

---

### Official Review · Reviewer_LucQ · 2023-06-25

**Soundness:** 2 fair
**Presentation:** 3 good
**Contribution:** 3 good
**Rating:** 6
**Confidence:** 3

**Summary:**

This paper considers a new setting of linear DAG learning problem. Based on a linear transform of linear SEM, authors propose to study a new setting where there are few "root causes", with potential measurement noise in the data. Identifiability is proved and the true DAG is shown to be the global minimizer of the L0-norm of the vector of "root causes", under a specific distribution on the "root cause" variables.

**Strengths:**

- a new setting for the linear DAG learning problem
- useful identification result (Thm. 3.2) with a complete proof



**Weaknesses:**

- the new setting and its motivating example are not sufficiently convincing.
- authors only consider specific distribution on the "root causes" variables, making  theoretic result somewhat limited
- some results are trivial from the literature (e.g., Thm 2.1)

**Questions:**

- My first concern is about the new setting of learning linear DAGs; it is not clear whether the new setting is indeed meaningful in practice. In the pollution model example, it is stated that "the relevant DAG data is triggered by sparse events on the input size and not by random noise ", and "We assume a DAG describing a river network. The acyclicity is guaranteed since flows only occur downstream. .. We assume that the cities can pollute the rivers." In this example, why do we need to learn DAGs? The graph structure can be more accurately obtained by getting the information of flows. As such, I suggest authors give more practical examples in the context of DAG learning, to make the new setting indeed meaningful.

- root causes: in the DAG learning literature, "root causes" generally refer to the source nodes of DAGs. Not sure if it is suitable to use a (somewhat) conventional name to refer to something new in the same context.

* Theorem 2.1 is not new and may be not stated as a theorem.
* Regarding Thm 3.1: similarly, the result simple follows from the LiNGAM result, by assuming a specific distribution on the "root causes", so maybe consider put it as a lemma or proposition. Besides, in the experiments in the supplementary material, I can see LiNGAM failed. Can you explain why? After all, the linear SEM falls exactly into the setting of LiNGAM if there no measurement noise.
* after Eq. 8, "Among all possible DAG matrices, the solution of the optimization problem (8) is the one that minimizes the number of the root causes X": can you give more details about this claim?
* This may be a bit picky, but only sparse graphs (with edge/node=2 and 3) are considered. Please try other degrees of graphs. (But this is not very important and may be added after the rebuttal.)
* please use \cite, \citet properly; e.g., line 131 line 181-182

Overall, I like the new setting of learning linear DAGs, but every new setting should be validated with more examples/details. I look forward to author response.

**Limitations:**

Authors discussed a limitation that the proposed method only works for few root causes in the paper. To me, another important limitation is the specific distribution assumption on the "root causes", as the proof heavily depends on this assumption.

---

> ### Author Rebuttal · Authors · 2023-08-09
>
> # Response to reviewer LucQ
>
> ## Weaknesses
>
> **Motivation for root causes.** Please see our general reply for a better motivation and also our success in a causal discovery competition with real-world data.
>
> **Assumptions on the root causes.** Yes, within the scope of sparse root causes we only consider the special case that the support of the root causes is a multivariate Bernoulli variable (since the values are either zero or nonzero) and we should mention it in the limitations. We considered it a reasonable assumption to execute our ideas. Identifiability will hold for larger classes of distributions but we did not run experiments.
>
> **Theorem 2.1.** We will call it a lemma. It is indeed an easy consequence of basic linear algebra (and thus not novel of course) but usually not explicitly stated in papers on linear SEMs. We do so here since this input/output view of a linear SEM is fundamental to our work.
>
> ## Questions
>
> **Meaningfulness in practice.** Please see our general reply for a better motivation and also our success in a causal discovery competition with real-world data.
>
> **Term: Root causes.** Source nodes are sometimes called root nodes but the term root causes seems not common. In any event, we will better clarify better its use in our work to avoid ambiguity.
>
> **Theorem 2.1.** See reply above.
>
> **Theorem 3.1 and LiNGAM's performance.** Yes, Theorem 3.1. is a consequence of the input being non-Gaussian, which we also state. Better to call it a corollary to the LiNGAM result than a lemma.
>
> The subpar performance of LinGAM is an interesting question and a possible reasoning can be found in [Shimizu et.al, 2006], Section 5. Theoretically, the LiNGAM algorithm is guaranteed to find the true DAG. However, in practice, there exist small estimation errors. Moreover, exhaustive search over all possible permutations to compute the true causal order is infeasible for a large number of nodes. Thus, it turns out that the algorithm approximates with zeros the smallest elements and then computes the corresponding permutation to make it upper triangular. This makes the algorithm approximate and it is unclear whether it is guaranteed to find the true matrix in practice. For example, as we see in our experiments, it can fail.
>
> **Optimization Objective, after Eq. (8).**
> For a DAG matrix $\widehat{\mathbf{A}}$ the quantity $\mathbf{X}\left(\mathbf{I} + \overline{\widehat{\mathbf{A}}}\right)^{-1}$ is equal to the root causes $\widehat{\mathbf{C}}$ that would generate the data $\mathbf{X}$ via equation (7). Therefore, the optimization objective in (8) minimizes the overall number of root causes (number of nonzero entries of $\mathbf{C}$) for the entire data matrix $\mathbf{X}$. We will write this better.
>
> **Edge density.** We chose to follow the experimental settings of prior work, which mostly generates sparse DAGs. For completeness, we conducted an experiment with varying average degree included it in the attached pdf, Fig. 2. For degree up to roughly half the maximal possible degree, we still perform best.
>
> **Citations.** Thanks, we will fix it.
>
> ## Limitations
>
> **Root causes distribution assumption.** Yes, we will mention in the limitations that we only considered multivariate Bernoulli. Identifiability will generalize to other distributions (due to the non-Gaussianity needed in Theorem 3.1) but we did not run any experiments with any. We did mention in the limitations that fixed support is not supported (and it seems also not identifiable).

---

> > ### Comment · Reviewer_LucQ · 2023-08-12
> > **Thanks for response**
> >
> > My major concern has been mostly resolved. I believe that authors can make the setting more practically convincing in future revision.
> >
> > An additional suggestion is to add a dicussion regarding LiNGAM's performance, as the thoery part depends on LiNGAM's. From my point of view, it is even better to have a more through investigation, e.g., some extra experiments.
> >
> > I will increase my score accordingly.

---

### Official Review · Reviewer_STdi · 2023-07-06

**Soundness:** 3 good
**Presentation:** 2 fair
**Contribution:** 2 fair
**Rating:** 5
**Confidence:** 3

**Summary:**

This paper considers learning of linear SEMs (weight matrix) under a data generation process that differs from the common formulation. It is assumed that each sample is generated from only few number of non-zero noise variables in which the set of noise variables is stochastic. The main theoretical result is identifiability of the weight matrix via an L0 norm objective. A relaxation of the objective with an L1 norm loss is used for extensive synthetic experiments and shown to be effective for learning the weight matrix under the proposed data generation mechanism.

**Strengths:**

- Sparse root causes are an interesting concept to explore.
- Experiments on synthetic data demonstrate strong performance. Specifically, the proposed algorithm is fast and seems to scale up well, and when the assumptions are violated slightly, the algorithm still yields reasonable results.

**Weaknesses:**


- While the sparse root causes assumption is interesting, it is not well-motivated. For the pollution example, if I understand correctly, the top left and bottom left nodes do not really play “causal” roles in the sense that they are deterministic mediators in the system, and the whole causal system can be represented without using those nodes. In this sense, “few root nodes” become the effective causal nodes. Am I missing something? The numerical evaluations using real data also do not provide much help for motivation.
- NNZ is not a strong metric when SHD is already provided, and the only edge of the proposed algorithm in the real data experiments is this metric.
- Presentation can be improved. For instance, Theorem 2.1 formulation of the linear SEM is trivial. Similarly, having $N_c$ and $N_x$ separately is superfluous; having a single small variance, not necessarily isotropic noise suffices for the description.

**Questions:**


- The authors can elaborate on the Weaknesses item 1.

- An additional question for the presentation. In L103-105,  “The high-level intuition is that it can be reasonable to assume that, with the viewpoint of (3), the relevant DAG data is triggered by sparse events on the input size and not by random noise.”
At first, I thought that you only consider a fixed subset of the nodes that have “random noise” with large magnitudes at each sample. If that was the case, the data is not i.i.d, the events are sparse on the input size, but it doesn’t necessarily occur on the same set of variables and I was puzzled with the importance of the proposed mechanism. Then I realized in Theorem 3.1 that at every sample, the non-zero noise variables are chosen randomly and it made sense (please correct me if I misunderstood anything). For a paper that proposes a new data generation model, the presentation should have been cleaner.

- Theorem 3.2.. Given a large enough but finite number of samples, but not a sample complexity result. This is a rather weird statement.

- Note: figure 2 has Möbius as the algo name. Perhaps it’s forgotten in the main text.

**Limitations:**

The limitations of this work are discussed very clearly in Section 6. I thank the authors for that paragraph.

---

> ### Author Rebuttal · Authors · 2023-08-09
>
> # Response to reviewer STdi
>
> ## Weaknesses
>
> **Motivation for root causes.** Please see our general reply for a better motivation and also our success in a causal discovery competition with real-world data.
>
> **Root causes support.** Note that we assume varying support of the root causes in the dataset $\mathbf{X}$. The example in Fig. 1 shows only one data vector with a specific root cause support of size 2. Because of the varying support, the system cannot be simplified and all nodes are significant, possibly in distinct samples (data vectors).
>
> **NNZ metric.** We will adjust the text to only refer to SHD, SID as indicators for the performance of our algorithm. However, NNZ is valuable as an indicator that the output of the algorithms is not trivial. For example, the empty DAG would result in an SHD of 17 in Table 4 (since the true DAG only has 17 edges). We will clarify this in the text.
>
> **Anisotropic noise.** Yes, one can fuse the noise variables $\mathbf{N}_c$ and  $\mathbf{N}_x$ into a single (anisotropic) one:
> $$\mathbf{N} =\mathbf{N}_c + \mathbf{N}_x\left(\mathbf{I} - \mathbf{A}\right).$$
> However, we prefer to keep them separate since they have distinct and intuitive meanings in a real-world setting: $\mathbf{N}_c$ captures approximately (not exact) sparse root causes, $\mathbf{N}_x$ is the measurement noise always present when obtaining real data $\mathbf{X}$. Also, this allows us to manipulate them separately later in our experiments (Table 1, rows 9, 10).
>
> ## Questions
>
> **Fixed support of root causes.** Yes, we have to clarify earlier that the root causes have varying support. We only consider fixed support in one of the synthetic experiments (Table 1, row 13).
>
> **Sample complexity.** Indeed not well written, we will rewrite. In the supplementary material we provide a bound on the number of samples in the noise-free case. Empirically, our algorithm works with much less samples and in the presence of noise, which shows that this bound is very loose.
>
> **Möbius.** It is a mistake, we'll fix.

---

> > ### Comment · Reviewer_STdi · 2023-08-14
> >
> > I thank the authors for their rebuttal. My assessment of the paper remains the same.

---

### Official Review · Reviewer_NZKD · 2023-07-14

**Soundness:** 3 good
**Presentation:** 2 fair
**Contribution:** 2 fair
**Rating:** 5
**Confidence:** 2

**Summary:**

This paper presents a new formulation of linear SEM by specifying a structure on the noise variables, which seems to impose zero-inflated distrbutions to achieve the "few roots" modelling goal. Identifiablity is given and guarantee for $L^0$ minimization estimator is provided for a special noise-free case. The minimization problem is further formulated into continuous optimization and numeric experiments are conducted.

**Strengths:**

- The idea of the new formulation is interesting, based on the illustrative river pollution example.
- The experiments consider many different setups, also including real datasets. The proposed algorithm shows comparable performance in the simulation study, though not by much.

**Weaknesses:**

- The motivation of the proposed formulation needs more elaboration, see questions. And the current definitions of "few roots" and "negaligible noise" in (6) are not formal or clear.
- Thm 3.2 only works for noise-free setting, thus there is no guarantee for consistency of $L^0$ minimization estimator for general model (4) beyond noise-free.
- Neither the related work and experiments discuss or compare with constraint-based methods, even the basic PC algorithm. Is it becaus we lose Markov property and faithfulness in this new formulation?

**Questions:**

- Thm 3.1 is proved by transforming the proposed model into a non-Gaussian DAG model. Just want to clarify: is the proposed formulation covered by the orginal DAG model? If so, are all other DAG learning approaches applicable? In that case, how do the authors justify the superiority of the new formulation beyond the empirical experiments?
- In Figure 2(a) and (b), the SHD and SID for mobius are larger than others at the beginning but drop in the end, any explanation on that?

**Limitations:**

Lmitations are discussed in Section 6.

---

> ### Author Rebuttal · Authors · 2023-08-09
>
> # Response to reviewer NZKD
>
> ## Strengths
>
> **Improvement against other methods.** Our algorithm offers significant improvements over the baselines in most scenarios of the simulation experiment (Table 1 and Fig. 2) and especially when the number of nodes in the graph is scaled up (Table 3).
>
> ## Weaknesses
>
> **Motivation of our setting.** Please read our general reply for a better explanation of the motivation and also our success in a causal discovery competition on real-world data.
>
> **Assumption on the noise and root causes.** Our assumptions are formally defined in Eq. (6). In words, the root causes are approximately (as captured by $\mathbf{N}_c$) sparse and the data is (as always) subject to measurement noise $\left(\mathbf{N}_x\right)$, where both noise terms are of low magnitude as described in (6). In the concrete experiments we consider concrete numbers for the error energies. We will clarify.
>
> **Theorem 3.2 and the presence of noise.** Indeed, we have included this statement as limitation (b) in section 6.
>
> **Constraint-based methods.** We did not include constraint-based methods since prior work already has shown to outperform them. So we focused on the best baselines. However, we include a comparison against the PC algorithm in the attached pdf, Fig. 1. The PC algorithm performs inferior.
>
> ## Questions
>
> **Superiority of our formulation.** Yes, our model still assumes a linear SEM via Eq. (5). Thus, it differs only in the assumption on the distribution and structure of what was viewed before as noise, which we view now as the input provided by each node to produce the DAG data as output (Eq. (6)). In this new formulation we assume (approximately) sparse input and the optimization problem we solve captures that. Thus the proposed input-output viewpoint of the linear SEM and motivates the assumption of having sparse root causes.
>
> **Low data regime.** Yes, it appears that in the low data regime our method performs worse. This is likely due to the sparsity of the root causes (which have varying support by assumption): to identify all edges the root cause supports of the data vectors may need to cover a significant subset of the nodes.

---

> > ### Comment · Reviewer_NZKD · 2023-08-14
> >
> > I thanks the authors for the response, which addressed some of my concern. I have increased the score. I hope the authors make corresponding change in the revision, especially, make eq (6) more formal, e.g. the definition seems to be put on the realized data instead of data generating process, which is unconventional; and "significantly larger" is not rigorous math language.

---

### Author Rebuttal · Authors · 2023-08-09

# General Comments
We thank all the reviewers for their kind reviews and their effort and interest in understanding and commenting on our work. We will incorporate these comments in an improved revision. Here, we would like to address two main points that arose from the comments.

**Motivation for few root causes**
We view the pollution example as a metaphor that can be moved to various DAG scenarios as roughly sketched next.

For example, as pointed out by reviewer LucQ, the data could measure (amount of) water flow in which case the root causes would capture cities with major input (e.g., through rainfalls).

In gene networks that measure gene expression, few root causes would mean that few genes are activated in a considered dataset.

In a citation network where one measures the impact of keywords/ideas, few root causes would correspond to the few origins of them.

[Peters et. al., 2017] mention in pp. 19-21 the *Principle of independent mechanisms* which in essence makes the assumption that any causal data generation relies on independent mechanisms that turn the input into output. This corresponds to our viewpoint on the linear SEM with the independent root causes as input.

As a thought: with the input/output view of linear SEMs in Eq. (3) one can question why all real world DAG data should be generated from i.i.d. noise $\mathbf{N}$ as input.

**Application in real-world data**
Our method doesn't achieve the best performance on the dataset from [Sachs et. al., 2005], but is reasonably competitive, which is all we wanted to show.

*More importantly*, after the NeurIPS submission our method was among the three winning entries in a causal discovery competition, run by a major pharma company, with the results presented in a (non-archival) report and at a major ML conference (no specifics due to double-blind review). One reviewer commented "... the few root causes assumption may have biological relevance worthy of further investigation." Our method was the only entry that offered some theoretical guarantees and performed well even with a linearity assumption.

Also, motivated by a comment of Reviewer Aevs we did an experiment to estimate the root causes of the dataset from [Sachs et. al., 2005]. In the pdf, Fig. 4 we show the result. The dataset comes with an unweighted adjacency matrix. We chose weights to enforce sparsity on the root causes. The sparsity roughly holds (Fig. 4, right). Interestingly (Fig. 4, left), the root causes tend to have fixed support, which could explain why we are unable to perfectly recover the DAG.

---

### Decision · Program_Chairs · 2023-09-21

**Decision:**

Accept (poster)

**Comment:**

After discussion, there was a weak consensus to accept this paper. The authors provide a novel perspective on identifying DAGs via root causes, and while the results have some limitations (outlined in the reviews), proposing new directions for identifying causal graphs is an important direction for the community. The inclusion of additional experiments during the rebuttal was convincing for several reviewers. For this reason, I recommend accept.

Nonetheless, I hope the authors will take the author feedback into account carefully. In particular, the following:
- There is a lack of rigor in some places that can be tightened up
- Even though previous work may have shown that the PC algorithm has limitations, the authors should replicate this behavior on their own, and include the results in the appendix (I would consider including GES as well)